# Tubulin glutamylation regulates axon guidance via the selective tuning of microtubule-severing enzymes

Daniel Ten Martin[1], Nicolas Jardin[1], Juliette Vougny[2], François Giudicelli[3], Laïla Gasmi [1], Naomi Berbée [2,4], Véronique Henriot[5,6], Laura Lebrun [5,6], Cécile Haumaître [7], Matthias Kneussel [8], Xavier Nicol [2], Carsten Janke [5,6], Maria M Magiera [5,6✉], Jamilé Hazan [1,9,10✉] & Coralie Fassier [1,2,10✉]

## Abstract

The microtubule cytoskeleton is a major driving force of neuronal circuit development. Fine-tuned remodelling of this network by selective activation of microtubule-regulating proteins, including microtubule-severing enzymes, has emerged as a central process in neuronal wiring. Tubulin posttranslational modifications control both microtubule properties and the activities of their interacting proteins. However, whether and how tubulin posttranslational modifications may contribute to neuronal connectivity has not yet been addressed. Here we show that the microtubule-severing proteins p60-katanin and spastin play specific roles in axon guidance during zebrafish embryogenesis and identify a key role for tubulin polyglutamylation in their functional specificity. Furthermore, our work reveals that polyglutamylases with undistinguishable activities in vitro, TTLL6 and TTLL11, play exclusive roles in motor circuit wiring by selectively tuning p60-katanin- and spastin-driven motor axon guidance. We confirm the selectivity of TTLL11 towards spastin regulation in mouse cortical neurons and establish its relevance in preventing axonal degeneration triggered by spastin haploinsufficiency. Our work thus provides mechanistic insight into the control of microtubule-driven neuronal development and homeostasis and opens new avenues for developing therapeutic strategies in spastin-associated hereditary spastic paraplegia.

**Keywords** Axon Guidance; Spastin; Katanin; Tubulin Polyglutamylation; Hereditary Spastic Paraplegia
**Subject Categories** Cell Adhesion, Polarity & Cytoskeleton; Neuroscience; Post-translational Modifications & Proteolysis

## Introduction

Over the past decades, microtubules (MTs) have emerged as key players in nervous system development. In addition to providing mechanical support, MTs form railways for axonal transport and mediate key signalling events that contribute to neuron morphological and behavioural changes during neural circuit wiring (Conde and Caceres, 2009; Hoogenraad and Bradke, 2009; Dent et al, 2011). Numerous studies have established MT remodelling as a key driving force in axon outgrowth and guidance (Zang et al, 2021; Atkins et al, 2023). Notably, the asymmetric invasion and stabilisation of MTs within the growth cone filopodia drive directed axon outgrowth (Williamson et al, 1996; Zhou et al, 2002; Buck and Zheng, 2002). Moreover, several guidance cues influence MT dynamics within the growth cone and thereby growth directionality in vitro, and a growing number of MT-interacting proteins are found to be required for neuronal circuit wiring in vivo (Sánchez-Huertas and Herrera, 2021; Atkins et al, 2023). While these data illustrate the central role of MTs in shaping neuronal connectivity, the intrinsic mechanisms underpinning the fine-tuned remodelling of this network in developing axons are far from being understood.

Importantly, tubulin posttranslational modifications (PTMs) have the potential to generate subpopulations of MTs that exhibit distinct physical properties and may locally fulfil specific cellular functions (Janke and Magiera 2020; Katrukha et al, 2021; McKenna et al, 2023). Tubulin polyglutamylation, which consists in the addition of glutamate side chains to tubulin C-terminal tails, is an abundant tubulin PTM in neurons (Audebert et al, 1994). This modification is catalysed by enzymes from the Tubulin Tyrosine Ligase-Like family (TTLL; Janke et al, 2005; van Dijk et al, 2007). Each TTLL has an enzymatic specificity generating either short or long chains of glutamates, and a substrate preference for α- versus β-tubulin (van Dijk et al, 2007). Lately, tubulin polyglutamylation was shown to fine-tune the binding affinity and/or activity of several MT-interacting proteins (Lacroix et al, 2010; Valenstein and Roll-Mecak, 2016; Genova et al, 2023) and the

[1]Sorbonne Université, INSERM U1130, CNRS UMR8246, Neuroscience Paris Seine - Institut de Biologie Paris-Seine (NPS-IBPS), Paris, France. [2]Sorbonne Université, CNRS, Inserm, Institut de la Vision, F-75012 Paris, France. [3]Institut de Biologie de l'École Normale Supérieure, ENS, CNRS UMR8197, INSERM U1024, Paris, France. [4]Amsterdam University Medical Center (UMC), University of Amsterdam (UvA), Amsterdam, The Netherlands. [5]Institut Curie, Université PSL, CNRS UMR3348, Orsay, France. [6]Université Paris-Saclay, CNRS UMR3348, Orsay, France. [7]Université Paris Diderot, INSERM UMR1149, ERL CNRS 8252, Paris, France. [8]Department of Molecular Neurobiology, Center for Molecular Neurobiology, ZMNH, University Medical Center Hamburg-Eppendorf, Hamburg, Germany. [9]Center for Interdisciplinary Research in Biology, Collège de France, CNRS UMR7241, INSERM U1050, Paris, France. [10]These authors contributed equally: Jamilé Hazan, Coralie Fassier. ✉E-mail: maria.magiera@curie.fr; jamile.hazan@college-de-france.fr; coralie.fassier@inserm.fr

dysregulation of its physiological levels impairs neuronal homeostasis in mice and humans (Magiera et al, 2018; Shashi et al, 2018). Yet, the physiological role of this tubulin PTM, as well as the selectivity of TTLL-generated polyglutamylation patterns towards MT-interacting proteins in developing neurons, remain poorly characterised.

Interestingly, TTLL-mediated tubulin polyglutamylation promotes MT severing by spastin and p60-katanin in vitro (Lacroix et al, 2010; Valenstein and Roll-Mecak, 2016; Shin et al, 2019; Genova et al, 2023). Notably, these critical regulators of MT mass and organisation (McNally and Roll-Mecak, 2018; Vemu et al, 2018; Kuo et al, 2019) are central for nervous system development and homeostasis (Lynn et al, 2021; Costa and Sousa, 2022). Indeed, these MT severers are involved in human cortical malformations for p60-katanin (Eom et al, 2014; Hu et al, 2014; Mishra-Gorur et al, 2014) and hereditary spastic paraplegia for spastin (Hazan et al, 1999). During development, both spastin and katanin play a decisive role in vertebrate axon outgrowth and branching (Ahmad et al, 1999; Karabay et al, 2004; Yu et al, 2008; Butler et al, 2010), while only spastin has been involved in axon guidance to date (Jardin et al, 2018).

We here explore whether TTLL-mediated MT polyglutamylation may constitute an original mechanism controlling neuronal circuit wiring via the selective regulation of severingenzyme activities. By combining loss-of-function and rescue experiments in zebrafish larvae, we first identify p60-katanin as a novel key player in motor axon navigation and characterise its non-redundant role with spastin in this process. We next provide the first in vivo evidence that long-chain glutamylases TTLL6 and TTLL11 selectively tune p60-katanin- and spastin-mediated motor axon targeting during development. We further provide a proof of concept that MT polyglutamylation generated by specific TTLL enzymes rescues the axonal phenotypes caused by defective spastin gene dosage both in zebrafish larvae and mammalian neurons. Overall, our work pinpoints tubulin polyglutamylation as a central regulator of MT functions in the developing brain as well as an appealing therapeutic target for neurological disorders involving MT-severing enzymes.

## Results

### p60-Katanin controls zebrafish spinal motor axon pathfinding and larval locomotion

The established role of the MT-severing protein Spastin in zebrafish motor circuit wiring (Jardin et al, 2018) prompted us to explore its functional similarity/diversity with another MT severer present in developing axons, p60-Katanin (Karabay et al, 2004), via loss-of-function analyses. Using *in toto* immunohistochemistry and in vivo live-imaging approaches in Tg(*Hb9*:GFP) 72-hours post-fertilisation (hpf) transgenic larvae expressing GFP in motor neurons, we showed that *p60-katanin* knockdown affected secondary motor axon (sMN) targeting in a dose-dependent manner (Fig. 1A–E). Indeed, while low doses of p60-Katanin morpholino (MO$^{p60Kat/1.3\,pmol}$) led to the abnormal split and targeting of motor dorsal nerves (empty arrows, Fig. 1A,B; Movie EV1), its higher doses (MO$^{p60Kat/3.4\,pmol}$) significantly impaired dorsal nerve formation (asterisks, Fig. 1A,C) compared to MO$^{Ctl}$ and MO$^{p60Kat/1.3\,pmol}$ embryos (full arrows for MO$^{Ctl}$, Fig. 1A,C). Further-more, MO$^{p60Kat/3.4\,pmol}$ morphants and to a lesser extent MO$^{p60Kat/1.3\,pmol}$ larvae also exhibited misrouted rostral nerves that were aberrantly

targeted caudally (empty arrowheads, Fig. 1A,D). This phenotype was barely observed in control larvae (full arrowheads, Fig. 1A,D; Movie EV2). Moreover, 57% of MO$^{p60Kat/3.4\,pmol}$ larvae showed sMN axons that exited the spinal cord at ectopic sorting points (red arrowhead, Fig. 1A,E), compared to 10% in MO$^{p60Kat/1.3\,pmol}$ and 3% in MO$^{Ctl}$ larvae. Notably, the sMN defects of MO$^{p60Kat/1.3\,pmol}$ and MO$^{p60Kat/3.4\,pmol}$ morphant larvae were rescued by injecting proportional doses of human *KATNA1* mRNA (Fig. 1A–E) and were also observed following the injection of another *p60-katanin* morpholino (MO$^{katna1aug1}$; Butler et al, 2010; Fig. EV1A–C). These results confirm that these sMN phenotypes were specifically due to a dose-dependent lack of p60-Katanin.

To further validate the specificity of the *p60-katanin* morphant phenotype, we analysed the development of sMN axons in the *katna1$^{sa18250}$* mutant (hereafter called *katna1* mutant; Fig. 1F–I), harbouring a point mutation (G > A) in the donor splice site of *katna1* intron 4 (Fig. EV2A). This mutation caused aberrant insertions of intron-4 fragments into *katna1* transcripts, which all led to a frameshift and the appearance of a premature STOP codon depleting a large portion of the protein, including the ATPase domain (Fig. EV2B–D). Notably, both zygotic (*katna1$^{-/-Z}$*) and maternal zygotic (*katna1$^{-/-MZ}$*) *katna1* mutants displayed significant sMN axon pathfinding errors, including the abnormal split of the dorsal nerves and less frequently, the caudal misrouting of the rostral nerves (Fig. 1F–I). These anomalies were similar to those described for *p60-katanin* morphants and were rescued by the overexpression of human *KATNA1* mRNA (Fig. 1F–I), strengthening their phenotypic specificity. Furthermore, the increased frequency of dorsal and rostral nerve defects in *katna1$^{-/-Z}$* compared to *katna$^{+/-}$* larva (Fig. 1G–I) confirmed that p60-Katanin gene dosage was critical for motor axon targeting. Altogether, these data demonstrate that p60-Katanin controls vertebrate motor axon navigation, in addition to its well-established role in axon extension.

The axon pathfinding defects of *p60-katanin* morphants and mutants (*katna1$^{-/-MZ}$*) were associated with a curved-tail pheno-type (Movie EV3 for morphants and Fig. EV2E,F for mutants) as well as a striking locomotor deficit in the touch-evoked escape response test at 72 hpf. These motility defects were characterised by reduced swimming speed and covered distances compared to control larvae (Fig. 2A–F; Movie EV3) and were rescued with adequate doses of human *KATNA1* mRNA (Fig. 2A–F; Movie EV3). These data were consistent with the rescue of the sMN axon guidance defects (Fig. 1) and confirmed that these locomotor phenotypes could be specifically assigned to p60-Katanin deficiency. Remarkably, although less severely affected, *katna1* mutants recapitulated the morphological, behavioural and axon pathfinding defects of *p60-katanin* morphants. These results ruled out the possibility of non-specific effects due to morpholino injection or chemical ENU mutagenesis. RT-qPCR analysis of p60-Katanin-related protein-encoding transcripts revealed a sig-nificant increase in the expression levels of two MT-severing enzymes, Spastin (*spast*), and Fidgetin (*fign*), as well as a *katna1* paralogue, Katanin-like 2 (*katnal2*) in *katna1$^{-/-MZ}$* embryos compared to controls (Fig. EV2G). This analysis suggests that genetic compensation by some *katna1*-related genes (Monroe and Hill, 2016) may account for the milder phenotype of *katna1* null mutants. Altogether, our data establish a specific role for p60-Katanin in zebrafish motor circuit wiring and larval locomotor behaviour.

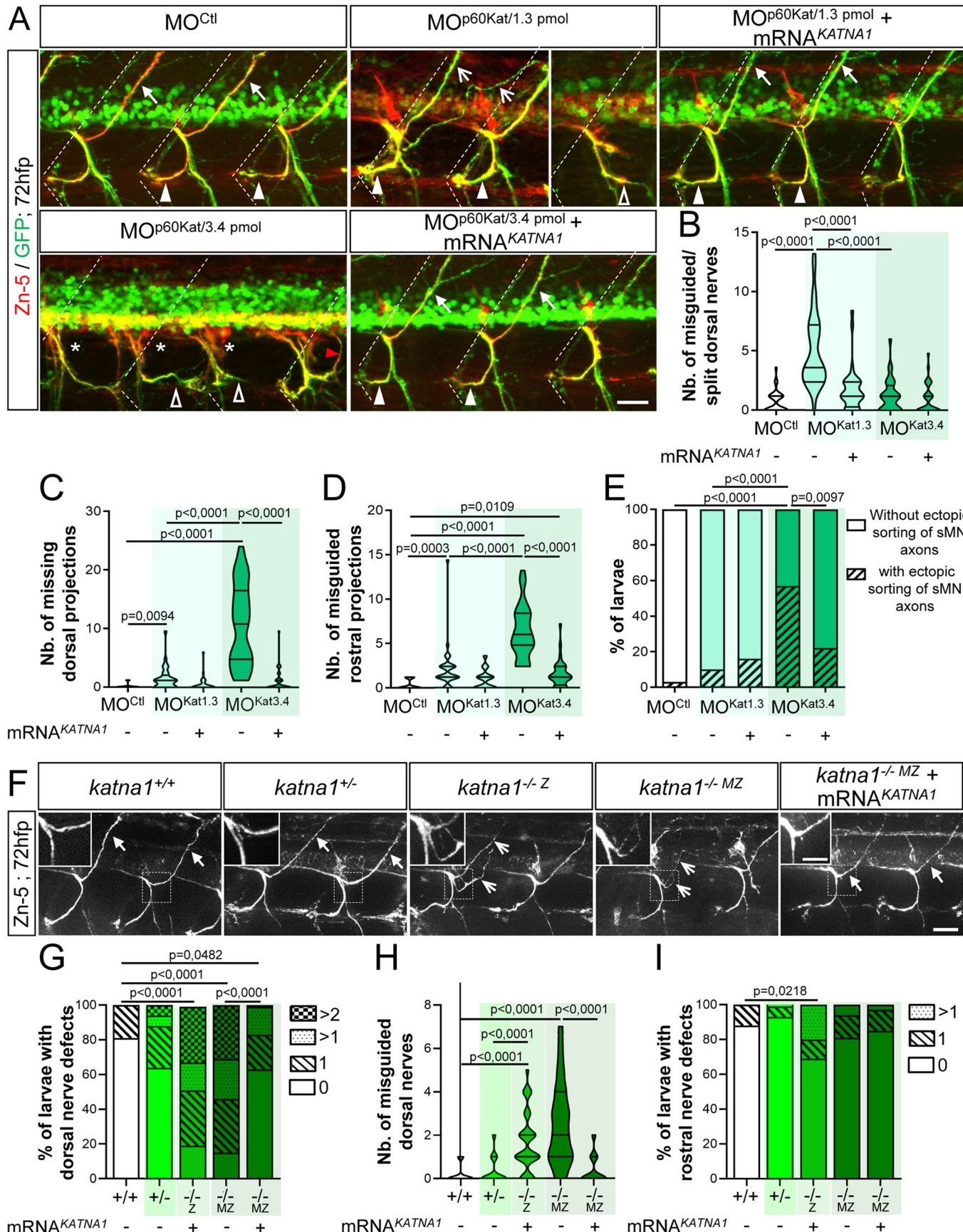

**Figure 1. *p60-katanin* impairs the axon pathfinding of secondary motor neurons.**

(A) Immunolabelling of sMN tracts in 72-hpf Tg(*Hb9*:GFP) larvae injected with control morpholino (MO^Ctl; $n = 32$), *p60-katanin* morpholino (MO^p60Kat/1.3pmol, $n = 32$ and MO^p60Kat/3.4pmol, $n = 32$) or co-injected with *p60-katanin* morpholino and human *KATNA1* mRNA (MO^p60Kat/1.3pmol + mRNA^KATNA1, $n = 32$; MO^p60Kat/3.4pmol + mRNA^KATNA1, $n = 32$) using Zn-5 and GFP antibodies. Dotted lines mark lateral myosepta. (B–D) Mean number of misguided dorsal projections (B), missing dorsal projections (C) and misguided rostral nerves (D) per larva. (E) Percentage of larvae with ectopic sorting of sMN axons from the spinal cord. (F) Immunolabelling of sMN axons in 72-hpf controls (*katna1^+/+^*, $n = 34$), as well as heterozygous (*katna1^+/-^*, $n = 53$), homozygous zygotic (*katna1^−/−Z^*, $n = 33$) and maternal zygotic (*katna1^−/−MZ^*, $n = 12$) mutant larvae using Zn-5 antibody. Insets are higher magnifications of the dorsal nerve. (G) Percentage of larvae with dorsal nerve defects. (H) Mean number of misguided dorsal projections. (I) Percentage of larvae with rostral nerve defects. (A, F) Lateral views of the trunk, anterior to the left. Full arrowheads and full arrows show normal rostral and dorsal nerves, respectively. Empty arrowheads and empty arrows indicate misguided rostral and split/misguided dorsal projections, respectively. Asterisks indicate missing dorsal nerves while the red arrowhead points at aberrant exit points of sMN axons from the spinal cord. Scale bars: 25 μm, inset: 10 μm (F). (B–E, G–I) Non-blind quantifications were performed on 24 spinal hemisegments around the yolk tube per larva. Analysed larvae were pooled from three independent experiments. (B–D, H) Violin Plots; horizontal bars indicate the median ± the 1st and 3rd quartiles. Kruskal–Wallis ANOVA test with Dunn's post hoc test. (E, G, I) Chi² test. *P* values are displayed on graphs. Source data are available online for this figure.

## P60-Katanin and Spastin play non-overlapping roles in motor circuit wiring

Zebrafish p60-Katanin and Spastin were suggested to have distinct, although related functions in primary motor neuron axon outgrowth (Butler et al, 2010). To support this paradigm, we showed that the depletion of p60-Katanin or Spastin in zebrafish embryos caused eminently different sMN defects, including the abnormal split of the dorsal nerve (p60-Katanin, Fig. 1A,B) or the ectopic sorting of motor neuron somata from the spinal cord (Spastin, Jardin et al, 2018). However, both morphants/mutants also exhibited overlapping sMN axon pathfinding defects (e.g., the aberrant caudal targeting of the rostral nerve; Fig. 1A,D and Jardin et al, 2018) that could reflect partial functional redundancy. To assess the functional specificity of these closely related MT-severing enzymes in motor neuron development, we conducted cross-rescue analyses. We showed that ubiquitous overexpression of human spastin (i.e., injection of 200 pg/embryo of human spastin-encoding *SPAST/SPG4* mRNA) failed to rescue the sMN axon pathfinding defects of MO^p60Kat/1.3pmol morphant (Fig. 3A–C) unlike human p60-katanin overexpression (Fig. 1). Moreover, the same dose of human *SPAST/SPG4* mRNA did not affect sMN axon targeting in larvae injected with a control morpholino (Fig. 3B,C) whereas it efficiently rescued the sMN axon pathfinding errors of zebrafish *spastin* morphants (Jardin et al, 2018). Conversely, ubiquitous overexpression of human p60-katanin (using the same dose of *KATNA1* mRNA as above, see Fig. 1) failed to alleviate the axon targeting defects of Spastin-depleted sMN axons (Fig. 3D–F), while no gain-of-function phenotype of sMN axon development was observed in MO^CTL-injected larvae (Fig. 3D–F). These data demonstrate that the microtubule-severing enzymes p60-Katanin and Spastin play non-overlapping key roles in vertebrate motor axon navigation.

## The knockdown of tubulin glutamylases TTLL6 and TTLL11 affects motor axon targeting in a similar way to p60-Katanin and Spastin loss of function

Despite their similar expression pattern in the zebrafish developing spinal cord (Fig. EV3A for *p60-katanin* and Arribat et al, 2020 for *spastin*) and their co-expression in the majority of spinal motor neurons (Fig. EV3B), the MT severers p60-Katanin and Spastin have non-redundant functions in motor circuit wiring (Fig. 3). This finding prompted us to assess whether the functional specificity of these two enzymes could be underlain by their selective preference

for distinct populations of MTs. We focused our analysis on tubulin polyglutamylation, since this posttranslational modification is highly abundant in neurons (Janke and Kneussel, 2010) and was shown to promote spastin- and katanin-severing activity in vitro (Sharma et al, 2007; Lacroix et al, 2010; Valenstein and Roll-Mecak, 2016; Shin et al, 2019; Genova et al, 2023). Tubulin polyglutamylation is catalysed by Tubulin Tyrosine Ligase-Like (TTLL) enzymes, whose role in neuronal circuit wiring has never been addressed. The enrichment of *ttll6* and *ttll11* transcripts in the zebrafish developing spinal cord at stages of motor axon outgrowth (Pathak et al, 2011) incited us to examine whether these two long-chain-generating glutamylases could be involved in this developmental process. We first confirmed the presence of polyglutamylated MTs in zebrafish spinal motor axons using two antibodies recognising either long and short chains of glutamates (GT335) or long chains only (polyE). Glutamylated MTs were detected in both pMN axons (left panels, 26 hpf, Fig. 4A) and sMN nerve tracts (right panels, 72 hpf, Fig. 4A), including the dorsal and rostral nerves, which pathfinding was respectively affected by p60-Katanin and Spastin loss of function. We next used a morpholino-based knockdown approach to assess whether TTLL6 and TTLL11 depletion (MO^TTLL6 or MO^TTLL11) impaired sMN axon targeting. Since TTLL11 loss of function impaired chromosome segregation fidelity in the zebrafish and led to early embryonic lethality (Zadra et al, 2022), we injected low doses (0.8 pmol) of MO^TTLL11 morpholinos in two-cell-stage embryos to partly knockdown TTLL11 expression and by-pass this mortality issue. To assess the impact of each glutamylase knockdown on MT polyglutamylation levels, we measured and compared the mean fluorescence intensity of PolyE signal along individual motor axons (i.e., the pioneer CaP axons) of 26-hpf control, MO^TTLL6- and MO^TTLL11-injected embryos. This strategy was adopted to avoid measurement biases at a nerve scale, which could arise from reduced axon contingency associated with morphant motor nerve guidance and defasciculation defects. We showed that the depletion of each glutamylase significantly reduced the level of long glutamate chains in pMN CaP axons (Fig. 4B,C), confirming TTLL6/TTLL11 knockdown efficiency and the key contribution of these two glutamylases to MT polyglutamylation in motor neurons.

At 72 hpf, while MO^TTLL6- and MO^TTLL11-injected larvae were indistinguishable in terms of gross morphology with a ventrally curved body axis characteristic of ciliary mutants (Fig. 4D, upper panels), both morphants showed substantially different sMN axon guidance phenotypes (Fig. 4D, middle and bottom panels). TTLL6-depleted larvae exhibited a significant number of split and

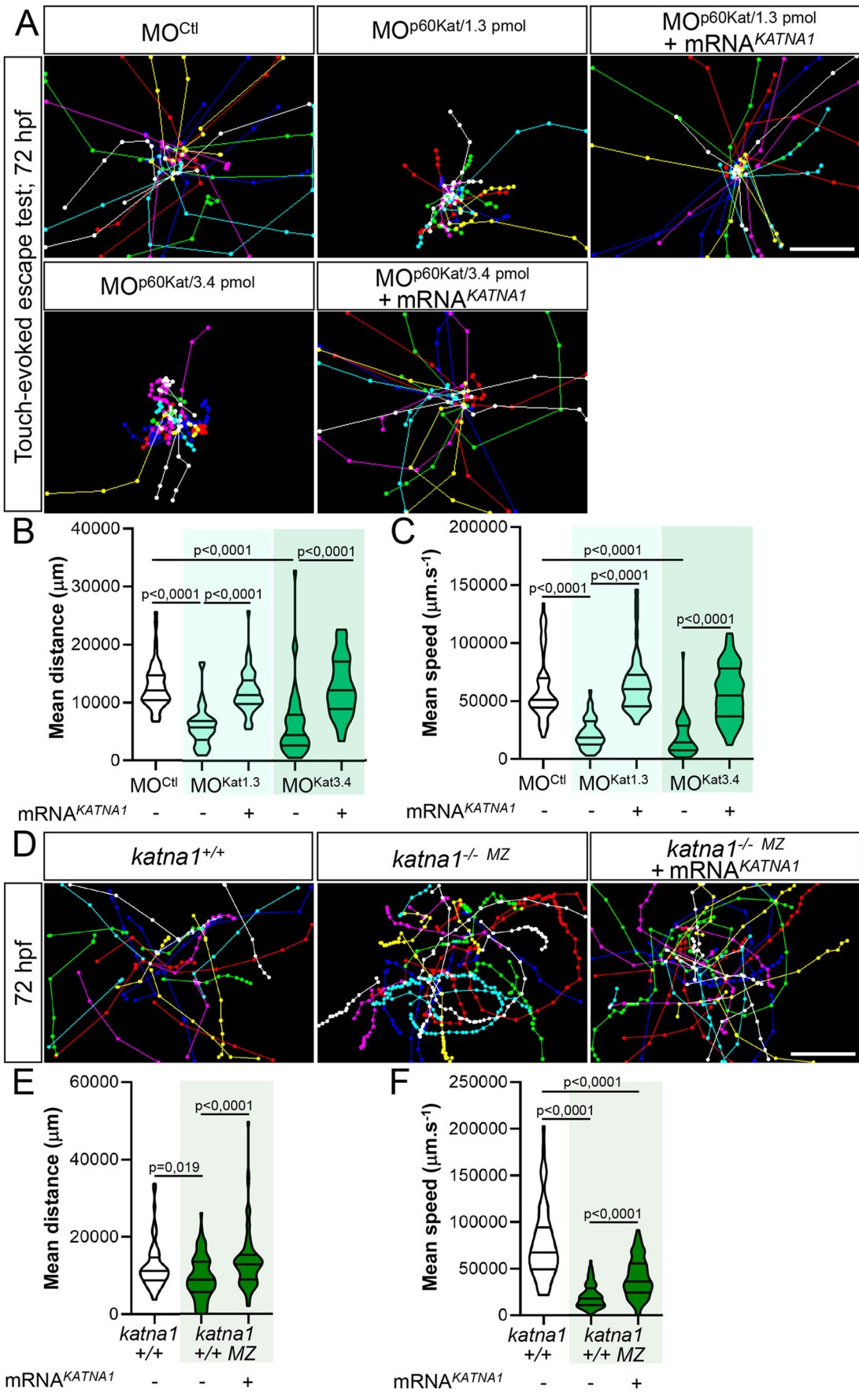

◄ **Figure 2. Loss of p60-Katanin causes a dramatic decrease in zebrafish larval mobility.**

(A) Touch-evoked escape behaviour of 72-hpf larvae injected with a control morpholino (MO$^{Ctl}$; $n = 90$), increasing doses of *p60-katanin* morpholino (MO$^{p60Kat/1.3pmol}$; $n = 102$ and MO$^{p60Kat/3.4 pmol}$; $n = 85$), or co-injected with MO$^{p60Kat}$ morpholino and human *KATNA1* mRNA (MO$^{p60Kat/1.3pmol}$ + mRNA$^{KATNA1}$, $n = 91$; MO$^{p60Kat/3.4pmol}$ + mRNA$^{KATNA1}$, $n = 97$). (B, C) Mean swimming distance (B) and speed (C) of the larvae tracked in (A). (D) Touch-evoked escape behaviour of 72-hpf control (*katna1*$^{+/+}$; $n = 65$) and *katna1* mutant larvae injected or not with human *KATNA1* mRNA (*katna1*$^{-/- MZ}$; $n = 74$ and *katna1*$^{-/- MZ}$ + mRNA$^{KATNA1}$; $n = 78$) and pooled from three independent experiments. (E, F) Mean swimming distance (E) and speed (F) of the larvae tracked in (D). (A, D) Each line represents the trajectory of one larva after touch stimulation while the distance between two dots indicates the distance covered by a larva between two consecutive frames. Scale bar: 5 mm. (B, C, E, F) Violin Plots; horizontal bars indicate the median ± the 1st and 3rd quartiles. Kruskal–Wallis ANOVA test with Dunn's post hoc test. P-values are displayed on graphs. Source data are available online for this figure.

misguided dorsal nerves (empty arrows, Fig. 4D,E) compared to control and *TTLL11* morphant larvae (full arrows, Fig. 4D,E), as shown for *p60-Katanin* mutants/morphants (MO$^{p60Kat/1.3pmol}$; Fig. 1A,B,F–H). In contrast, numerous dorsal tracts were missing in *TTLL11* morphants compared to *TTLL6* morphants and controls (Fig. 4D,F). Furthermore, rostral nerves of TTLL11 morphant were mostly defasciculated or missing (bracket, Fig 4D,G) while some SMN somata exited the spinal cord (yellow arrows, Fig 4D,H), two phenotypes reported for *spastin* mutants (Jardin et al, 2018) while rarely observed in *TTLL6* morphant and control larvae (Fig. 4D,G,H). Moreover, as shown for spastin- and to a lesser extent p60-Katanin-depleted larvae, both *TTLL6* and *TTLL11* morphants exhibited sMN rostral projections that were aberrantly targeted caudally (25% and 19%, respectively Fig. 4D,I), as well as sMN axons exiting the spinal cord at ectopic sorting points (asterisks, Fig. 4D,J).

Although partially overlapping, the motor neuron phenotypes of *TTLL6* and *TTLL11* morphants suggested that these two as yet indistinguishable polyglutamylases could have different functions in motor circuit wiring. Furthermore, the significant phenotypic similarities caused by *TTLL6* or *TTLL11* knockdown and p60-Katanin or Spastin depletion hinted at the functional relationship between tubulin glutamylation and MT severing in vertebrate motor axon targeting.

## TTLL6 and TTLL11 have non-redundant functions in motor axon navigation

To assess the specificity of *TTLL6* and *TTLL11* morphant phenotypes and clarify the functional redundancy or specificity of these related enzymes, we carried out rescue and cross-rescue experiments. Notably, overexpression of mouse TTLL6 but not TTLL11 significantly alleviated the morphological and sMN axon guidance errors of *TTLL6* morphants (Fig. 5A,C–H). Reciprocally, overexpression of TTLL11 substantially rescued the morphological and sMN axon pathfinding defects of *TTLL11* morphants, unlike TTLL6 overexpression (Fig. 5B,C–H).

Overall, these analyses reveal that these two glutamylases displaying biochemically similar activities in vitro play non-redundant key roles during zebrafish motor circuit development in vivo.

## Only TTLL6-mediated microtubule polyglutamylation rescues the axon pathfinding and larval locomotor deficit of *p60-katanin* morphants

The overlapping axon pathfinding phenotypes between *TTLL* morphants and p60-Katanin or Spastin-depleted larvae led us to hypothesize that these glutamylases could selectively regulate the

activity of these two microtubule-severing enzymes during zebrafish motor circuit wiring. To test this hypothesis, we assessed whether promoting TTLL6- or TTLL11-mediated tubulin polyglutamylation could attenuate or rescue the motor axon pathfinding and locomotor deficit caused by p60-Katanin partial depletion by boosting its residual activity. Given that the penetrance of heterozygous *katna1*$^{+/-}$ axon guidance defects is too low to conduct reliable rescue experiments (Fig. 1G–I), we carried out the rescue analyses of the *p60-katanin* morphant phenotype at the lower dose of morpholino, which however led to robust axon pathfinding defects of the dorsal nerves (MO$^{p60Kat/1.3 pmol}$; Fig. 1A,B). We showed that *TTLL6* but not *TTLL11* overexpression significantly rescued the sMN axon guidance defects (Fig. 6A–C) and locomotor deficit (Fig. 6A,D,E) of 72-hpf MO$^{p60Kat/1.3pmol}$-injected larvae compared to controls. Notably, a catalytic-dead variant of TTLL6 (TTLL6 dead; Fig. EV4A,B) failed to rescue *p60-katanin* morphant defects. Furthermore, MO$^{p60Kat/1.3 pmol}$ morphant embryos also exhibited a distal hyperbranching of primary motor (pMN) axons at 26 hpf (Fig. EV5A,B), a phenotype equally observed in TTLL6-depleted larvae (Fig. EV5C,D). Interestingly, this hyperbranching could be rescued in both cases by the overexpression of TTLL6 but not by TTLL11 (Fig. EV5C–G). These results thus suggest that TTLL6-mediated tubulin glutamylation selectively regulates p60-Katanin-driven motor axon pathfinding and larval locomotion.

## TTLL6 selectively boosts p60-Katanin activity in motor axons to control their targeting

To test whether the rescue of MO$^{p60Kat/1.3pmol}$ morphant phenotypes by TTLL6 overexpression specifically relied on TTLL6-mediated stimulation of the residual activity of p60-Katanin or whether it involved other MT regulatory proteins (e.g., Spastin), we reproduced the same rescue experiments in the *p60-katanin* null mutant background (*katna1*$^{-/-MZ}$) completely lacking p60-Katanin. TTLL6 overexpression (as TTLL11 overexpression) here failed to rescue the axon pathfinding errors and swimming deficit of *katna1*$^{-/-MZ}$ mutants (Fig. 6F–I). These findings, which strikingly contrast the significant impact of TTLL6 overexpression on morphants with residual Katanin levels (MO$^{p60Kat/1.3pmol}$), strongly suggest that TTLL6 selectively regulates p60-Katanin-mediated motor axon targeting. To further unravel the cellular mechanisms underlying the control of p60-Katanin function by TTLL6 in developing motor axons, we assessed the influence of TTLL6 on p60-Katanin-mediated MT-severing activity. To this end, we used the *UAS/GAL4* system to drive the mosaic expression of the MT plus-end tracking protein EB3 fused to GFP in spinal motor neurons of control, MO$^{p60Kat/1.3pmol}$ and TTLL6-injected MO$^{p60Kat/1.3pmol}$ larvae, and monitored MT plus-end dynamics in dorsally projecting sMN

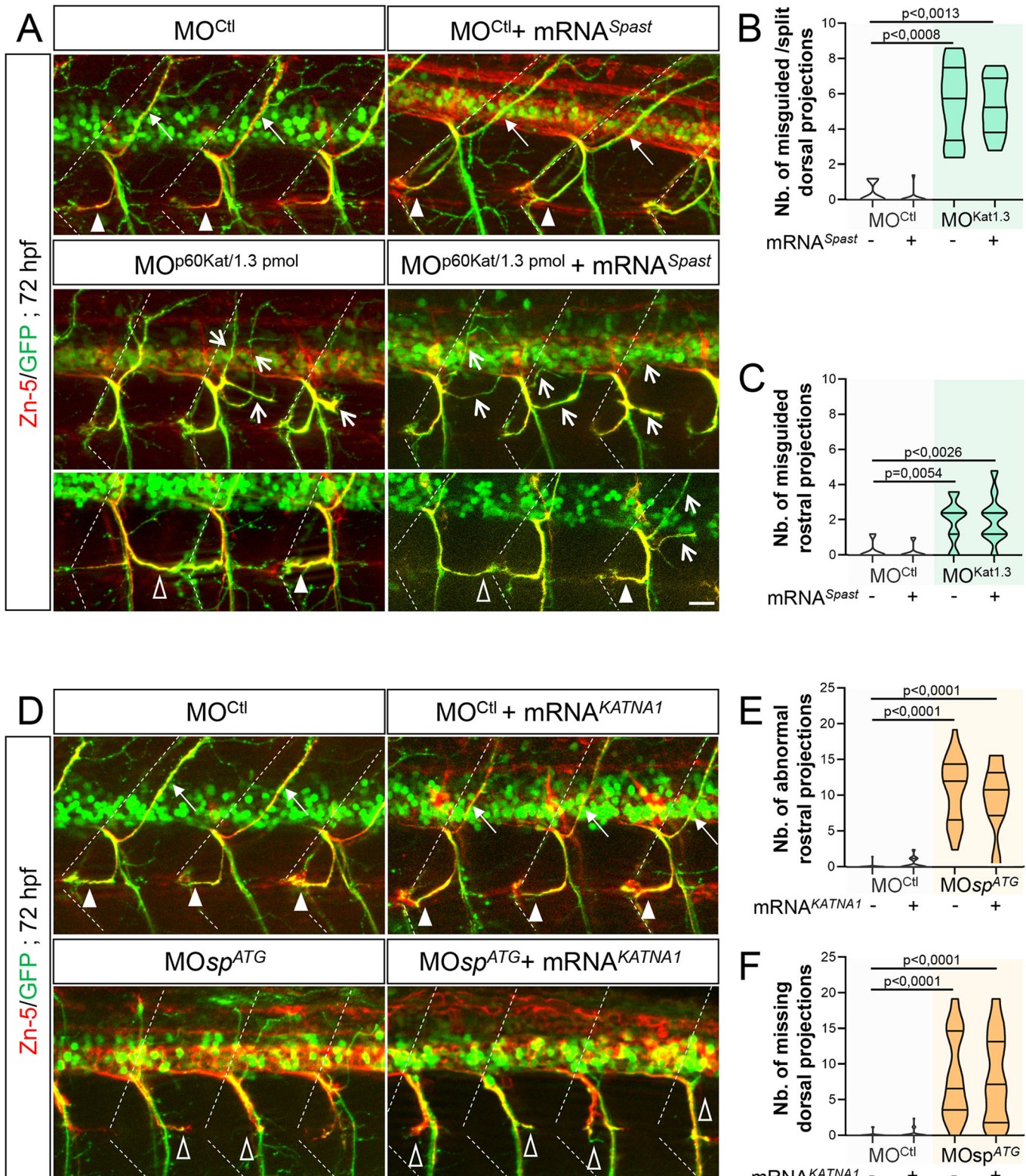

axons at a single axon scale. This allowed to show that *p60-katanin* knockdown reduced both the density (Fig. 7A–C) and growth speed (Fig. 7A,D; Movie EV4) of growing MT plus ends in these axons. Notably, these MT defects, which were consistent with a lack of MT-severing activity as previously described for p60-Katanin- and/ or Spastin-depleted axons (Butler et al, 2010; Fassier et al, 2013), were both rescued by the overexpression of TTLL6 (Fig. 7A–D; and Movie EV4), which strongly suggests that TTLL6 selectively boosts p60-Katanin activity in navigating motor axons to control their targeting.

**Figure 3. Spastin and p60-katanin have non-redundant roles in motor axon guidance.**

(A) Immunolabelling of sMN axons in 72-hpf Tg(*Hb9*:GFP) larvae injected with MO$^{Ctl}$ ($n = 10$), 1.3 pmol of MO$^{p60Kat}$ ($n = 10$), or co-injected with MO$^{Ctl}$ or MO$^{p60Kat}$ and human *SPG4* mRNA (MO$^{Ctl}$ + mRNA$^{Spast}$, $n = 14$; MO$^{p60Kat/1.3 pmol}$ + mRNA$^{Spast}$, $n = 10$) using Zn-5 and GFP antibodies. (B, C) Mean number of misguided/split dorsal nerves (B) and misguided rostral nerves (C) per larva. (D) Immunolabelling of sMN axons in 72-hpf Tg(*Hb9*:GFP) larvae injected with MO$^{Ctl}$ ($n = 29$), MO$sp^{ATG}$ ($n = 30$), or co-injected with MO$^{Ctl}$ or MO$sp^{ATG}$ and human *KATNA1* mRNA (MO$^{Ctl}$ + mRNA$^{KATNA1}$, $n = 32$; MO$sp^{ATG}$ + mRNA$^{KATNA1}$, $n = 29$) using Zn-5 and GFP antibodies. (E, F) Mean number of abnormal rostral nerves (i.e., caudally targeted or missing; E) and missing dorsal nerves (F) per larva. (A, D) Dotted lines indicate lateral myosepta. Full arrowheads and full arrows indicate normal rostral and dorsal nerves, respectively. Empty arrowheads and empty arrows indicate misguided rostral and dorsal projections, respectively. All images are lateral views of the trunk, anterior to the left. Scale bars: 25 μm. (B, C, E, F) Non-blind quantifications were performed on 24 spinal hemisegments around the yolk tube per larva. Analysed larvae were pooled from three independent experiments. Violin Plots; horizontal bars indicate the median ± the 1st and 3rd quartiles. Kruskal–Wallis ANOVA test with Dunn's post hoc test. *P* values are displayed on graphs. Source data are available online for this figure.

## TTLL11 selectively tunes spastin-driven motor axon navigation

To further characterise the functional selectivity of these long-chain tubulin glutamylases towards MT-severing enzymes in vivo, we undertook the same sets of rescue experiments in *spastin* morphant (knockdown) versus mutant (knockout) larvae. In contrast to our results on p60-Katanin, we now showed that TTLL11 but not TTLL6 overexpression significantly alleviated the axon pathfinding defects of rostral motor nerves caused by spastin partial knockdown (injection of 0.2 pmol of MO$sp^{ATG}$ morpholino; Fig. 8A,B). This selective improvement of *spastin* morphant axon trajectory by TTLL11 was associated with a slight but significant rescue of larval swimming speed compared to control or *TTLL6*-overexpressing *spastin* morphant larvae (Fig. 8A,C,D). Again, the overexpression of a catalytic-dead TTLL11 (TTLL11 dead) did not rescue the axon pathfinding defects of *spastin* morphant larvae (Fig. EV4C,D). Whether TTLL11 also regulates Spastin activity in pMN axons could not be assessed, since the partial knockdown of spastin (using 0.2 pmol of MO$sp^{ATG}$) did not affect pMN axon development. Overall, these data suggest that the guidance errors of sMN axons were not secondary to pioneering pMN axon pathfinding defects.

To assess whether the beneficial effect of TTLL11 on *spastin* morphant phenotype relied on TTLL11-driven enhancement of spastin activity or involved other MT-severing proteins, we next overexpressed TTLL11 or TTLL6 in the *spastin* CRISPR/Cas9 *null* mutant (*sp$^{C68X/C68X}$*; Jardin et al, 2018). Unsurprisingly, TTLL11 overexpression failed to rescue the rostral motor axon defects of *spastin* null mutants (Fig. 8E–G), identifying TTLL11 as a selective regulator of spastin-driven motor axon guidance.

Altogether, our results provide the first in vivo proof of concept that selective MT polyglutamylation patterns mediated by specific tubulin glutamylases rescue the motor neuron and locomotor defects caused by reduced levels of microtubule-severing enzymes in vivo.

## TTLL11 selectively rescues the axon degeneration hallmarks caused by spastin haploinsufficiency in a mammalian cellular model of hereditary spastic paraplegia

Importantly, mutations in the *SPG4* gene encoding the MT-severing enzyme spastin are responsible for the major form of autosomal dominant hereditary spastic paraplegia (HSP), a degenerative condition of the corticospinal tracts (Hazan et al,

1999). While few *SPG4* mutations were shown to have a toxic gain-of-function impact in overexpression experiments (Solowska et al, 2014; Qiang et al, 2019), the vast majority of *SPG4* mutations were reported to act through haploinsufficiency (Tarrade et al, 2006; Depienne et al, 2007; Roll-Mecak and Vale, 2008; Kasher et al, 2009; Fassier et al, 2013; Denton et al, 2014; Connell et al, 2016). Thus, increasing spastin activity above the critical threshold may be a relevant therapeutic strategy in *SPG4*-linked HSP cases.

Our identification of TTLL11 as a selective regulator of spastin activity in zebrafish developing motor neurons prompted us to carry out pilot experiments to assess whether TTLL11 overexpression may alleviate/rescue the axonal degeneration hallmarks caused by spastin haploinsufficiency in cultured cortical neurons from spastin heterozygous mouse mutant embryos (Sp + /-; Lopes et al, 2020). We here focused our analysis on axonal swellings, the most significant, easily quantifiable and evolutionarily conserved degenerative hallmark resulting from defective MT dynamics in spastin-deficient neurons (Tarrade et al; 2006; Kasher et al, 2009; Denton et al, 2014; Havlicek et al, 2014).

Yet, while axonal swellings of Sp + /- neurons exhibit the same characteristic features as those described in Sp-/- neurons, their number after 6 days in vitro (DIV6) was much lower than that of homozygous mutant neurons, which challenged our rescue analysis (Fig. EV6A–D and Tarrade et al, 2006). To overcome this difficulty, we performed a longitudinal analysis of primary cultures of Sp + /+ and Sp + /- cortical neurons. We showed that the percentage of axonal swellings significantly increased over time in vitro (from DIV6 to DIV9). We further established that DIV9 Sp + /- cortical neurons exhibited a significantly larger number of axonal swellings compared to controls, which was compatible with robust statistical analyses for rescue experiments (Fig. EV6A,C). Cultured Sp + /- neurons were thus transduced at DIV2 (before the occurrence of the swellings) with a lentivirus encoding wild-type or catalytically dead (as a negative control) human TTLL11 or mouse TTLL6 glutamylases fused to the GFP, and analysed at DIV9. For each condition, the percentage of swollen axons (Fig. 9A,B) and transduced neurons were evaluated at DIV9 on fixed neurons immunolabelled with anti-β-III tubulin and anti-GFP antibodies. We showed that wild-type TTLL11 rescued the axonal swelling phenotype caused by spastin haploinsufficiency, unlike TTLL6 or their respective catalytically dead variants (Fig. 9A,B), despite similar transduction efficiency (at least 97% for each experimental condition). Notably, TTLL11 overexpression reduced the percentage of axonal swellings in Sp + /- cultures to a level equivalent to that of control neuron cultures (Fig. 9B). Importantly, this beneficial effect of TTLL11 on the axonal swelling hallmark of

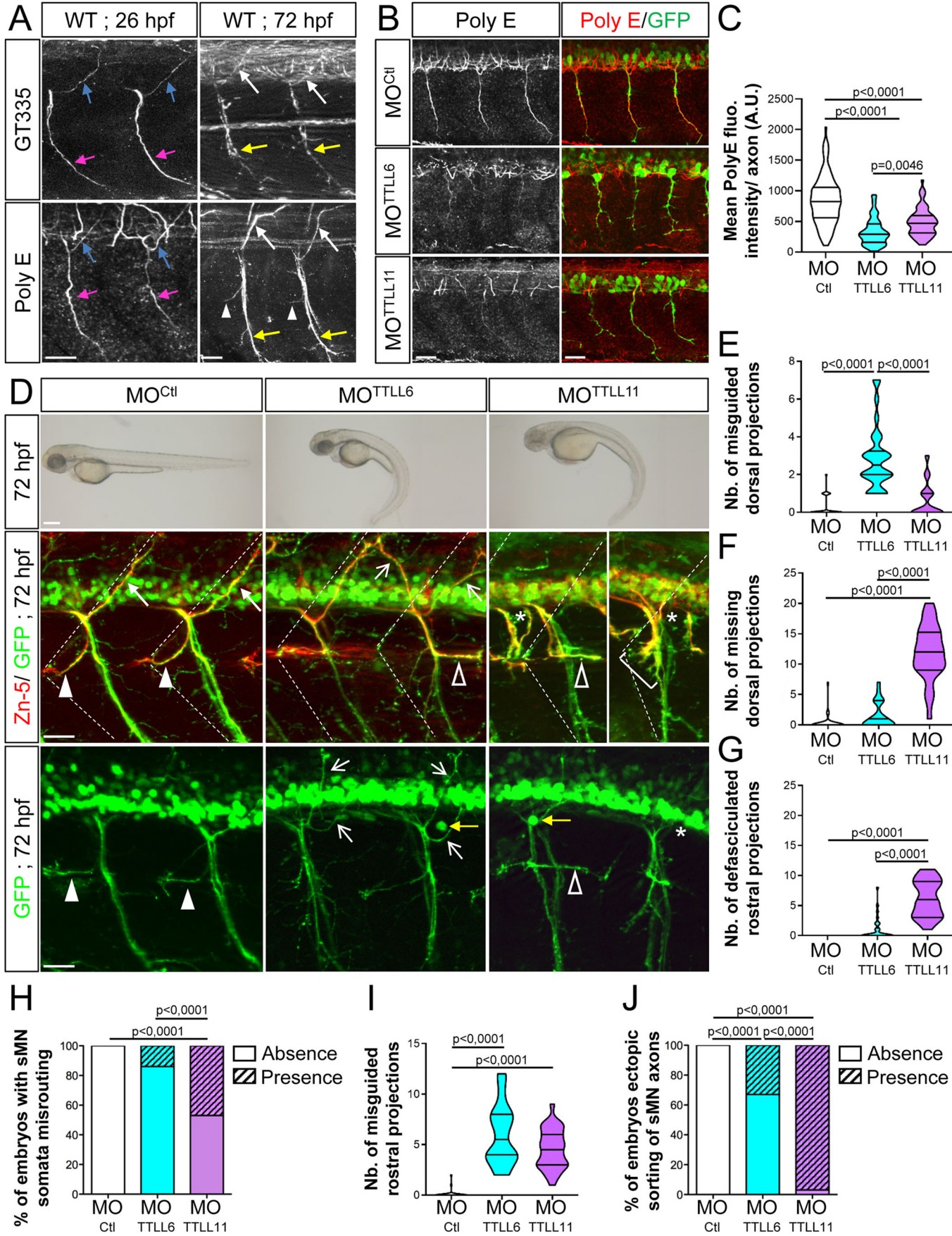

**Figure 4.   *TTLL6* and *TTLL11* knockdown leads to different motor axon pathfinding defects mimicking the respective phenotypes of p60-Katanin- and Spastin-depleted larvae.**

(A) Immunolabelling of polyglutamylated microtubules (MTs) in 26- (left panels) and 72-hpf (right panels) wild-type embryos using GT335 (upper panels) and polyE (lower panels) antibodies. Polyglutamylated MTs are observed in both pMN (26 hpf) and sMN (72 hpf) axons. Blue and pink arrows point at dorsally and ventrally projecting pMN axons, respectively. White arrows, yellow arrows and white arrowheads indicate dorsally, ventrally and rostrally projecting sMN axons, respectively. Scale bars: 25 μm. (B) Immunolabelling of polyglutamylated microtubules (MTs) in 26-hpf Tg(*Hb9*:GFP) embryos injected with MO^Ctl, MO^TTLL6 or MO^TTLL11 morpholinos using the polyE antibody. Scale bars: 25 μm. (C) Mean PolyE fluorescence intensity per axon (A.U.). Non-blind quantifications were conducted on MO^Ctl (*n* = 120), MO^TTLL6 (*n* = 74) or MO^TTLL11 (*n* = 68) pooled from two independent experiments. Twelve axons located around the yolk tube were analysed per embryo. (D) Upper panels: Overall morphology of 72-hpf control (MO^Ctl), *TTLL6* (MO^TTLL6) and *TTLL1* (MO^TTLL11) morphant larvae. Both *TTLL6* and *TTLL11* morphants exhibit a severe ventrally curved body axis phenotype compared to MO^Ctl-injected larvae. Scale bars: 250 μm. Middle and bottom panels: Immunolabelling of sMN axons in 72-hpf Tg(*Hb9*:GFP) larvae injected with MO^Ctl (*n* = 30), MO^TTLL6 (*n* = 30) or MO^TTLL11 (*n* = 30) larvae using Zn-5 and/or GFP antibodies. Dotted lines delineate lateral myosepta. Full arrowheads and full arrows, respectively, point at normal rostral and dorsal nerves. Empty arrowheads and empty arrows, respectively, indicate misguided rostral and dorsal projections. Brackets show defasciculated rostral nerves. Asterisks and yellow arrows, respectively, indicate the ectopic sorting of spinal motor neuron axons and somata from the spinal cord. Scale bars: 25 μm. (E–J) Quantifications of sMN defects in larvae analysed in (D). Mean number of split/misguided dorsal nerves (E), missing dorsal nerves (F), defasciculated/ missing rostral nerves (G) and misrouted rostral nerves (I) per larva. (H, J) Percentage of larvae with ectopic sorting of sMN somata (H) or axons (J) from the spinal cord. Non-blind quantifications were performed on 24 spinal hemisegments located around the yolk tube per larva. Analysed larvae were pooled from three independent experiments. (C, E–G, I) Violin Plots; horizontal bars indicate the median ± the 1st and 3rd quartiles. Kruskal–Wallis ANOVA test with Dunn's post hoc test. (H, J) Chi$^2$ test. *P* values are displayed on graphs. Source data are available online for this figure.

neuronal degeneration was completely lost when the same experiments were conducted in homozygous Sp-/- neurons (Fig. 9A,B), suggesting that TTLL11 may selectively tune spastin activity in mammalian cortical neurons. These data provide compelling evidence that enhanced TTLL11-mediated tubulin polyglutamylation can prevent (or delay) the axonal phenotypes caused by defective *spastin* gene dosage in animal and cellular models of *SPG4*-linked HSP, revealing an evolutionarily conserved mechanism (see Fig. 10A,B).

# Discussion

Among the myriad of molecules shown to play key roles in axon navigation, special focus has recently been given to MT regulatory proteins due to their prominent roles in integrating and translating extracellular guidance signals into changes in growth-cone mechanical behaviours (Lowery and Van Vactor, 2009; Dent et al, 2011; Kolodkin and Tessier-Lavigne, 2011).

The MT-severing enzyme p60-katanin is known to be a critical player in axon elongation (Ahmad et al, 1999; Karabay et al, 2004; Butler et al, 2010) and branching (Yu et al, 2008). Using loss-of-function and rescue approaches during zebrafish embryogenesis, we here unveil a key role for p60-Katanin in spinal motor axon navigation and targeting. Intriguingly, p60-Katanin regulates this developmental process in a dose-dependent manner, emphasising the precise fine-tuning of its activity in vivo. Interestingly, the orthologue of p60-katanin in *Drosophila* exerts a MT plus-end depolymerising activity in vitro in addition to its known severing activity (Zhang et al, 2011; Diaz-Valencia et al, 2011). The balance between these two functions seems to be influenced by its concentration, with the maximal depolymerisation rate occurring at lower concentrations than the maximal severing rate (Diaz-Valencia et al, 2011). This dual function of p60-katanin was reported to regulate MT dynamics at the actin cortex to fine-tune the migration of *Drosophila* S2 cells by suppressing protrusion (Zhang et al, 2011). Thus, if applicable in vivo in a vertebrate model, this concentration-dependent versatile role of p60-Katanin may provide an explanation for the dose-dependent effects of p60-Katanin depletion on axon targeting.

Furthermore, we have here demonstrated that p60-Katanin and its related protein Spastin have non-overlapping key roles in zebrafish motor axon navigation. This is consistent with their differential regulation of MT remodelling in non-neuronal cells (Lacroix et al, 2010; Shin et al, 2019) and axon branch formation in cultured hippocampal neurons (Yu et al, 2008). The functional specificity of these MT severers in axon branching has so far been linked to their distinct subcellular distribution in neurons and the differential regulation of their activities by the MT-associated protein Tau, which more efficiently protects MTs from being severed by p60-katanin than by spastin (Qiang et al, 2006; Yu et al, 2008). While these mechanisms could also underlie the functional specificity of spastin and p60-katanin in axon navigation, our analysis provides compelling in vivo evidence for a central role of MT polyglutamylation in their specificity.

Indeed, we here demonstrate that TTLL6 and TTLL11 selectively tune the respective activities of p60-Katanin and Spastin in developing axons to control their targeting. Notably, the interactions between MTs and some MAPs, including the p60-katanin regulator Tau, are sensitive to tubulin polyglutamylation levels in vitro (Boucher et al, 1994; Genova et al, 2023), as shown for spastin and katanin MT-severing activities (Lacroix et al, 2010; Zempel et al, 2013; Valenstein and Roll-Mecak, 2016; Shin et al, 2019; Genova et al, 2023). Overall, the model that emerges from these observations is that MT polyglutamylation patterns generated by distinct polyglutamylases may selectively regulate the activity of MT-severing enzymes by concomitantly (i) promoting their catalytic activity (i.e., increasing their coupling with the tubulin C-terminal tail, hexamerization and ATPase activity; White et al, 2007; Valenstein and Roll-Mecak, 2016; Shin et al, 2019; Zehr et al, 2020) and (ii) influencing the MT-binding affinity and/or activity of their regulators, including Tau (Yu et al, 2008) and CRMPs (Ji et al, 2018). Moreover, beyond their molecular specificity, the characteristic axon pathfinding phenotypes caused by the depletion of p60-Katanin or Spastin may also be the readout of their involvement as downstream targets of different axon guidance pathways, as shown for spastin main isoforms (Jardin et al, 2018).

Tubulin PTMs, including glutamylation, have been identified as key mechanisms that generate subpopulations of MTs with distinct properties and specific cellular functionalities (Janke and Magiera,

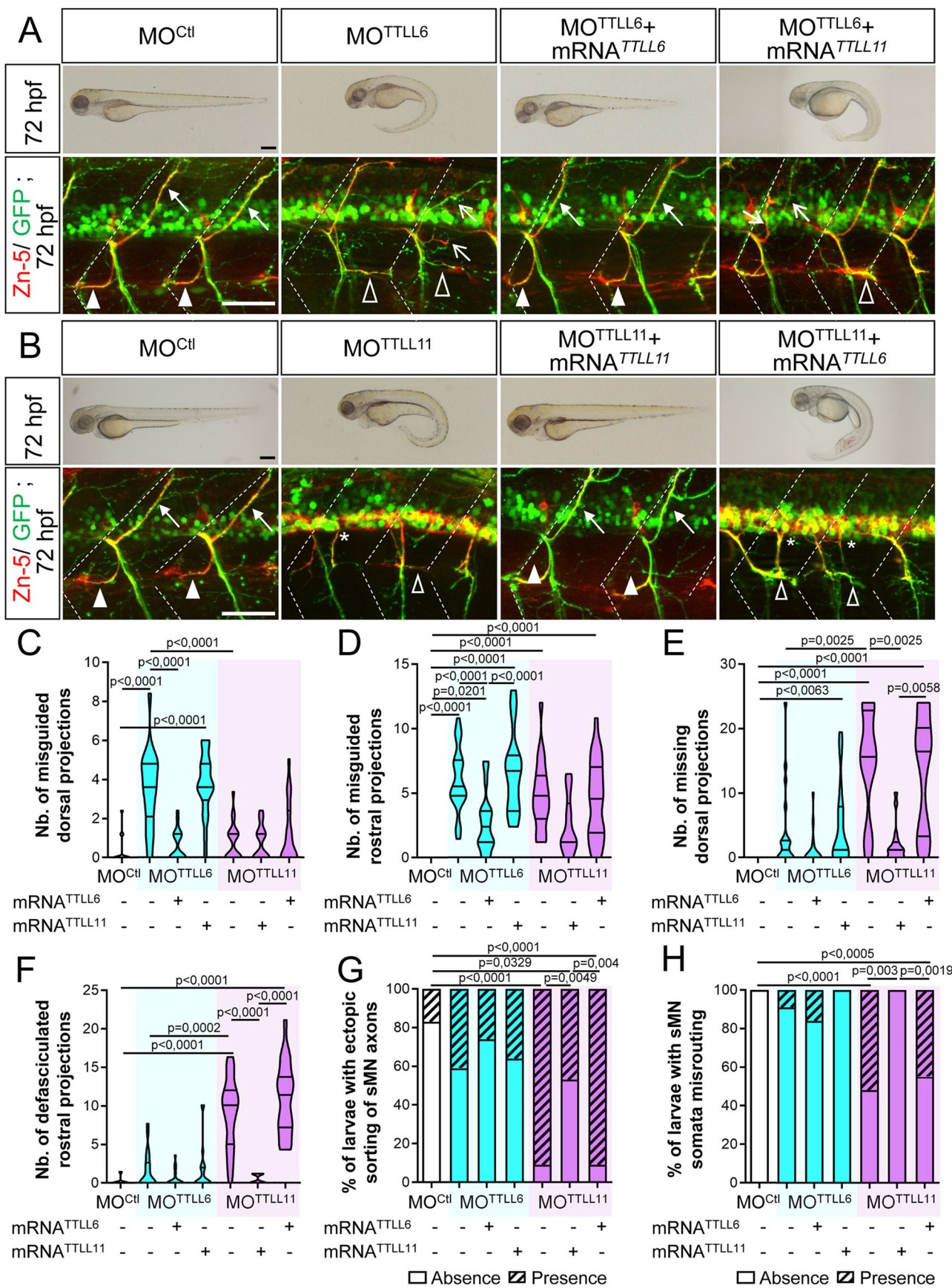

**Figure 5.   TTLL6 and TTLL11 play non-overlapping roles in spinal motor axon navigation.**

(A) Rescue and cross-rescue experiments of *TTLL6* morphant phenotypes by co-injection of mouse *TTLL6* or *TTLL11* transcripts. (B) Reciprocal rescue and cross-rescue experiments of *TTLL11* morphant phenotypes by co-injection of mouse *TTLL11* or *TTLL6* transcripts. (A, B) Upper panels: Overall morphology of 72-hpf control, morphant and rescued larvae. Scale bars: 250 µm. bottom panels: Immunolabelling of sMN axons with Zn-5 and GFP antibodies in 72-hpf Tg(*Hb9*:GFP) larvae injected with control (MO^Ctl, n = 22), *TTLL6* (MO^TTLL6, n = 22), *TTLL11* (MO^TTLL11, n = 21) morpholinos or co-injected with MO^TTLL6 or MO^TTLL11 and mouse *TTLL6* or *TTLL11* mRNA (MO^TTLL6 + mRNA^TTLL6, n = 19; MO^TTLL6 + mRNA^TTLL11, n = 22; MO^TTLL11 + mRNA^TTLL11, n = 17; MO^TTLL11 + mRNA^TTLL6, n = 22). Dotted lines delineate lateral myosepta. Full arrowheads and full arrows, respectively, indicate normal rostral and dorsal nerves. Empty arrowheads and empty arrows, respectively, point at misguided rostral and dorsal projections. Scale bars: 50 µm. (C–H) Quantifications of sMN defects in larvae analysed in (A, B). Mean number of split/misguided dorsal nerves (C), misrouted rostral nerves (D), missing dorsal nerves (E) and defasciculated rostral nerves (F) per larva. (G, H) Percentage of larvae with ectopic sorting of sMN axons (G) and somata (H) from the spinal cord. (C–H) Non-blind quantifications were performed on 24 spinal hemisegments located around the yolk tube per embryo or larva. Analysed larvae were pooled from three independent experiments. (C–F) Violin Plots; horizontal bars indicate the median ± the 1st and 3rd quartiles. One-way ANOVA test with Bonferroni's post hoc test (D) or Kruskal–Wallis ANOVA test with Dunn's post hoc test (C, E, F). (G, H) Chi$^2$ test. *P* values are displayed on graphs. Source data are available online for this figure.

2020; McKenna et al, 2023). For example, TTLL11 is required for spindle MT polyglutamylation in zebrafish embryos and its knockout impairs chromosome segregation fidelity and leads to early embryonic lethality (Zadra et al, 2022). Interestingly, we here showed that while the partial depletion of TTLL11 expression (using a morpholino-based knockdown strategy) does not lead to such devastating early developmental defects, it strikingly impairs neuronal circuit wiring, emphasising the extreme sensitivity of developing neurons to moderate changes in MT polyglutamylation levels.

Furthermore, based on their enzymatic specificity (addition of short or long glutamate side chains) and substrate preference (α- versus β-subunit of tubulin dimers), TTLL enzymes indeed generate different polyglutamylation patterns on MT outer surface (van Dijk et al, 2007). These distinct patterns were shown to fine-tune the activity of MT interactors including MT severers in vitro in a selective and gradual manner (Boucher et al, 1994; Lacroix et al, 2010; Valenstein and Roll-Mecak, 2016; Zheng et al, 2022; Genova et al, 2023). However, in vivo analyses confirming their crucial role in the selective regulation of MT interactors were still lacking until very recently. Excessive tubulin glutamylation due to carboxypeptidase (CCP) depletion was lately shown to impair axonal transport and cause neurodegeneration in mice and humans (Magiera et al, 2018; Shashi et al, 2018). Interestingly, these axonal transport defects and Purkinje cell degeneration caused by CCP1 depletion were revealed to be selectively rescued by the knockout of the α-tubulin glutamylase TTLL1 but not by the β-tubulin glutamylase TTLL7 knockout (Bodakuntla et al, 2021), thus establishing the relevance of TTLL catalytic specificities in neuronal homeostasis. Adding another layer of complexity to the tubulin code, our study shows that two long-chain generating tubulin glutamylases (TTLL6 and TTLL11) with similar substrate specificity (i.e., α-tubulin; van Dijk et al, 2007) are not functionally redundant and selectively boost the activity of different MT-severing enzymes in the wiring of vertebrate neuronal circuits underlying locomotion. Indeed, TTLL6 and TTLL11 may tune a specific MT severer by generating distinct MT glutamylation patterns (i.e., the number of added glutamate residues or the modification site on the tubulin tails). Identifying the glutamylation pattern specific of each TTLL and their impact on prime MT-dependent physiological processes will undoubtedly represent a major technological challenge for the years to come. Furthermore, our data together with previous studies exploring the tubulin tyrosination/detyrosination cycles in the mouse developing brain (Erck et al, 2005; Marcos et al, 2009; Pagnamenta et al, 2019) emphasise the urge to dissect the involvement of tubulin-modifying enzymes in physiological and pathological neuronal circuit wiring.

Finally, the rescue experiments in mammalian cortical neurons from spastin heterozygous mutant mice suggest that the selective capacity of TTLL11 versus TTLL6 to boost spastin activity in embryonic neurons is evolutionarily conserved. It further provides the proof of concept that selective modification of tubulin glutamylation patterns, herein through TTLL11 overexpression, rescues the degenerative hallmarks caused by spastin haploinsufficiency in a mouse cellular model of *SPG4*-linked HSP. This finding may pave the way towards the development of innovative MT-based therapeutic strategies in this major form of HSP. Moreover, together with a recent study showing that mutation of tubulin-alpha4a, which perturbs its polyglutamylation rescues the key pathological features of a humanised Tauopathy mouse model (Hausrat et al, 2022), our work pinpoints tubulin polyglutamylation as an attractive therapeutic target in MT-related neurodegenerative disorders (Rogowski et al, 2021).

In conclusion, we identify the specific roles of two in vitro biochemically indistinguishable tubulin polyglutamylases in axon guidance processes through the selective activation of defined microtubule-severing proteins. This allowed us to show how subtle variations in the tubulin code led to different functional readouts in vivo. Our findings also point at enzymes regulating tubulin polyglutamylation as potential therapeutic targets in pathological conditions associated with reduced levels of MT-severing enzymes.

# Methods

**Reagents and tools table**

| Reagent/resource | Reference or source | Identifier or catalogue number |
|---|---|---|
| **Experimental models** | | |
| Spastin KO mice (Sp-/-) | Kindly provided by the Kneussel lab; Lopes et al, 2020 | – |
| Tg(*Hb9*:GFP) fish | Kindly provided by the Houart Lab; (Flanagan-Steet et al, 2005) | Tg(*Hb9*:GFP) |
| Tg(*Mnx1*:GAL4) fish | Kindly provided by the Del Bene Lab; (Zelenchuk and Brusés, 2011) | Tg(*Mnx1*:GAL4) |

| Reagent/resource | Reference or source | Identifier or catalogue number |
|---|---|---|
| Zebrafish spastin CRISPR/Cas9 mutants | Nicol/Fassier lab (Jardin et al, 2018) | sp$^{C68X/C68X}$ |
| *Zebrafish p60-katanin* mutants (*katna1-/-*) | Sanger Centre; purchased from the ZIRC | *katna1$^{sa18250}$* |
| **Recombinant DNA** | | |
| Tol2-14*UAS*:EB3-GFP | Generated in the Nicol/ Fassier lab | – |
| pCS2 + _hum KATNA1 pCS2 + _hum spastin | This study | 'Methods' sections |
| pCS2 + _mTTLL11_dead pCS2 + _mTTLL11_WT pCS2 + _mTTLL6_dead pCS2 + _mTTLL6_WT | Kindly provided by the Janke lab (Lacroix et al, 2010) | |
| **Antibodies/probes** | | |
| Synaptotagmin 2 antibody | ZIRC, University of Oregon | Zn-5 |
| DM-GRASP/Neurolin antibody | ZIRC, University of Oregon | Znp-1 |
| GFP antibody (Rabbit) | Invitrogen (ThermoFisher Scientific, France) | A11122 |
| anti-Polyglutamylation Modification (GT335) | AdipoGen | GT335, no. AG-20B-0020B |
| anti-Polyglutamylation Modification (PolyE) | AdipoGen | polyE, no. AG-25B-0030-C050 |
| B-III tubulin | Biolegend | TUJ1; no. 801202 |
| sir-actin | Cytoskeleton | CY-SC001 |
| Rabbit GFP antibody | Torrey Pines Biolabs | TP-401 |
| Alexa Fluor 568 Goat anti-mouse antibody | Invitrogen (ThermoFisher Scientific, France) | A11019 |
| Alexa Fluor 555 Goat anti-mouse antibody | Invitrogen (ThermoFisher Scientific, France) | A-21422 |
| Alexa Fluor 488 Goat anti-rabbit antibody | Invitrogen (ThermoFisher Scientific, France) | A11008 |
| **Oligonucleotides and other sequence-based reagents** | | |
| PCR primers for cloning, genotyping and RT-PCR | This study | 'Methods' sections |
| RT-qPCR primers | This study | Table 1 |
| *katna1* sense and antisense probes | This study | 'Methods' sections |
| *spastin* and *p60-katanin* probes for hybridisation chain reaction (HCR) | Designed and synthesised by Molecular Instruments (Molecular Instruments Inc, Los Angeles, USA) | Accession numbers used for the design: NM_212915 and NM_001020604 |
| *p60-katanin, spastin, ttll6, ttll11* and control morpholinos | - This study - Developed by GeneTools (Philomath, USA) | 'Methods' sections |
| Human *SPAST/SPG4* and *KATNA1* mRNA, wild-type and catalytic-dead mouse *TTLL6* and *TTLL11* mRNA | - This study | 'Methods' sections |
| wild-type and catalytic-dead mouse TTLL6 and TTLL11 lentivirus | Kindly provided by the Janke lab and produced as described previously (Bodakuntla et al, 2020) | – |

| Reagent/resource | Reference or source | Identifier or catalogue number |
|---|---|---|
| **Chemicals, enzymes and other reagents** | | |
| HCR reagents | Molecular Instruments (Molecular Instruments Inc, Los Angeles, USA) | – |
| ProLong Gold mounting medium | Invitrogen (ThermoFisher Scientific, France) | P36934 |
| Restriction enzymes | New England Biolabs | |
| TOPO® TA cloning pcr4 vector | Invitrogen (ThermoFisher Scientific, France) | 450071 |
| SYBR Green master mix EurobioGreen qPCR Mix, Hi-ROX | Eurobio Scientific, France | GAEMMXOS1H-8T |
| SP6 mMessage mMachine kit | Ambion | AM1340 |
| SuperScript® III One-Step RT-PCR system | Invitrogen (ThermoFisher Scientific, France) | 12574030 |
| **Software** | | |
| Image J/ Fiji software | NIH | – |
| GraphPad Prism 9.00 | GraphPad Software, San Diego, CA | – |
| NIS-Elements Imaging Software | (Nikon Instruments Inc., Melville, USA) | – |
| MetaMorph software | Molecular Devices | – |
| **Other** | | |

## Zebrafish maintenance

Zebrafish embryos (*Danio rerio*) were obtained from natural spawning of Tg(*Hb9*:GFP) and Tg(*Mnx1*:GAL4) (Zelenchuk and Brusés, 2011) transgenic fish (Flanagan-Steet et al, 2005), *p60-katanin* mutants (*katna1$^{sa18250}$*; Sanger Centre, purchased from the ZIRC) or spastin CRISPR/Cas9 mutants (*sp$^{C68X/C68X}$*; Jardin et al, 2018). All embryos were maintained at 28 °C in E3 medium (5 mM NaCl/0.17 mM KCl/0.33 mM CaCl$_2$/0.33 mM MgSO$_4$/0.00001% (w/v) Methylene Blue) and staged by hours post-fertilisation (hpf) and gross morphology according to Kimmel et al (1995). To prevent pigment formation, 0.2 mM of 1-phenyl-2-thiourea (PTU, Sigma) was added to the E3 media from 24 hpf onwards.

All our experiments were made in agreement with the European Directive 210/63/EU on the protection of animals used for scientific purposes, and the French application 'Décret 2013-118'. The fish facilities of the Institut de Biologie Paris Seine and the Vision Institute have been approved by the French 'Service for animal protection and health', with the respective approval numbers A-75-05-25 and C-75-12-02.

## Morpholinos and RNA injections

Morpholino oligonucleotides (MO) blocking *p60-katanin/katna1, ttll6, ttll11* or *spastin* translation initiation sites as well as the standard control MO were developed by GeneTools (Philomath, USA) and designed as follows:

MO$^{ctl}$: 5′-CCTCTTACCTCAGTTACAATTTATA-3′,
MO$^{p60Kat}$: 5′-CTCATTGATCTCCCCCAAACTCATC-3′,

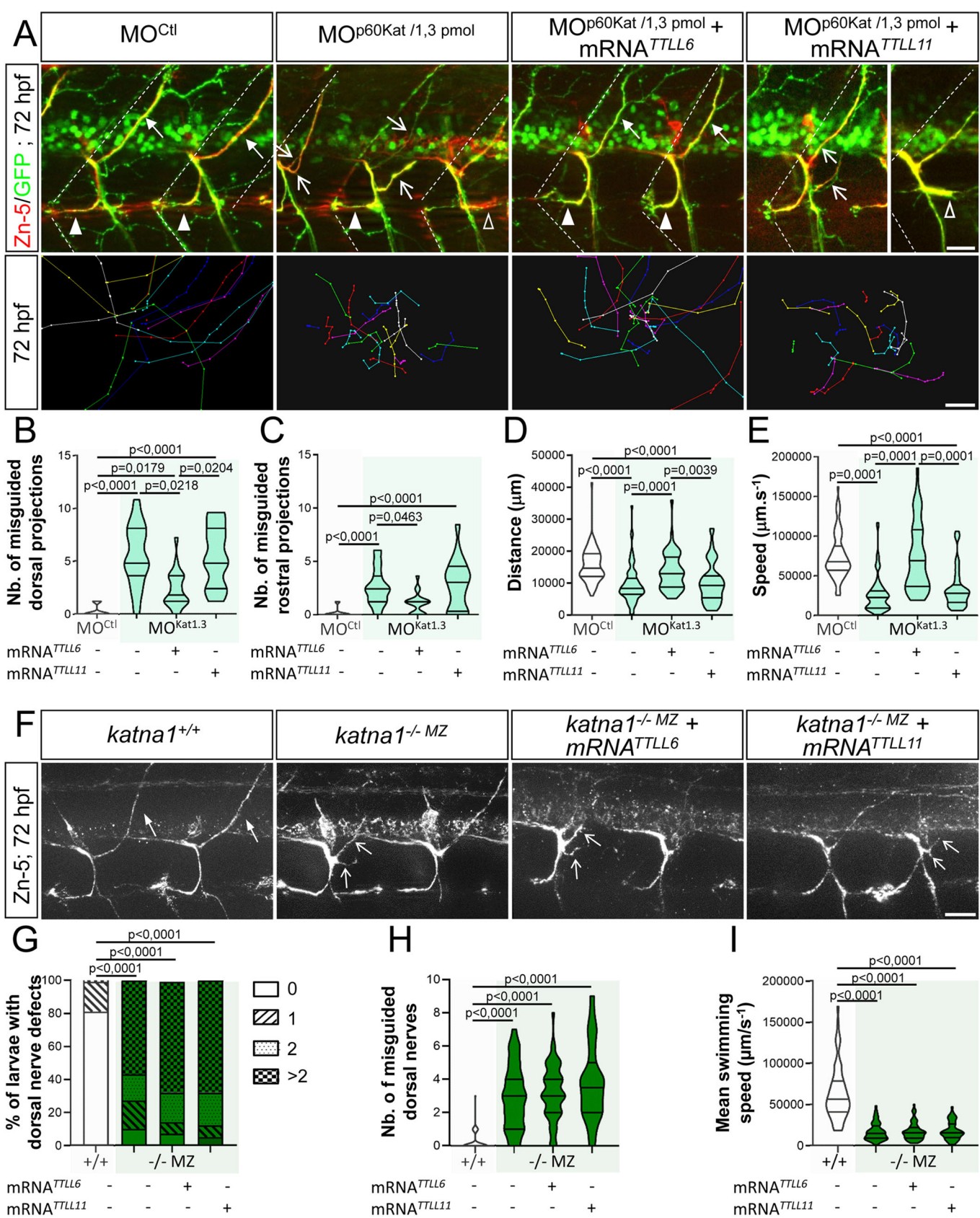

◄ **Figure 6. Selective regulation of p60-Katanin activity by TTLL6 is required for zebrafish motor axon targeting and larval locomotion.**

(A) Upper panels: Immunolabelling of sMN axon tracts using Zn-5 and GFP antibodies in 72-hpf Tg(*Hb9*:GFP) larvae injected with MO$^{Ctl}$ ($n = 20$), MO$^{p60Kat/1.3pmol}$ ($n = 20$) or co-injected with MO$^{p60Kat/1.3pmol}$ and mouse *TTLL6* ($n = 19$) or *TTLL11* ($n = 20$) mRNAs. Dotted lines delineate lateral myosepta. Lower panels: Tracking analysis of 72-hpf larvae injected with MO$^{Ctl}$ ($n = 30$), MO$^{p60Kat/1.3pmol}$ ($n = 30$) or co-injected with MO$^{p60Kat/1.3pmol}$ and mouse *TTLL6* ($n = 30$) or *TTLL11* ($n = 30$) mRNAs in a touch-evoked escape response test. Each line represents the trajectory of one larva after touch stimulation while the distance between two dots indicates the distance covered by a larva between two consecutive frames. Scale bar: 5 mm. (B–E) Quantifications of the sMN and locomotor defects of larvae analysed in (A) and pooled from three independent experiments. Mean number of split/misguided dorsal nerves (B) and misrouted rostral nerves (C) per larva. (D, E) Mean swimming covered distance (D) and speed (E). (F) Immunolabelling of sMN axon tracts with Zn-5 antibody in 72-hpf *katna1*$^{+/+}$ ($n = 57$), *katna1*$^{-/-MZ}$ ($n = 61$) and *katna1*$^{-/-MZ}$ larvae injected with mouse *TTLL6* ($n = 56$) or *TTLL11* ($n = 54$) mRNAs. (A, F) Full arrowheads and full arrows point at normal rostral and dorsal projections while empty arrowheads and empty arrows indicate misguided rostral and dorsal tracts, respectively. Scale bar: 25 μm. (G–I) Quantifications of dorsal nerve and motility defects of larvae analysed in (F) and pooled from two independent experiments. (G) Percentage of larvae with dorsal nerve defects. (H) Mean number of split/misguided dorsal nerves per larva. (I) Mean larval swimming speed in the escape-touch response test. Swimming speed values were extracted from tracking analysis of 72-hpf *katna1*$^{+/+}$ ($n = 65$), *katna1*$^{-/-MZ}$ ($n = 74$) and *katna1*$^{-/-MZ}$ larvae injected with the mRNAs encoding mouse TTLL6 ($n = 78$) or TTLL11 ($n = 47$). (B, C, G, H) Non-blind quantifications were performed on 24 spinal hemisegments located around the yolk tube per larva. (B–E, H, I) Violin Plots; horizontal bars indicate the median ± the 1st and 3rd quartiles. Kruskal–Wallis ANOVA test with Dunn's post hoc test. (G) Chi$^2$ test. *P* values are displayed on graphs. Source data are available online for this figure.

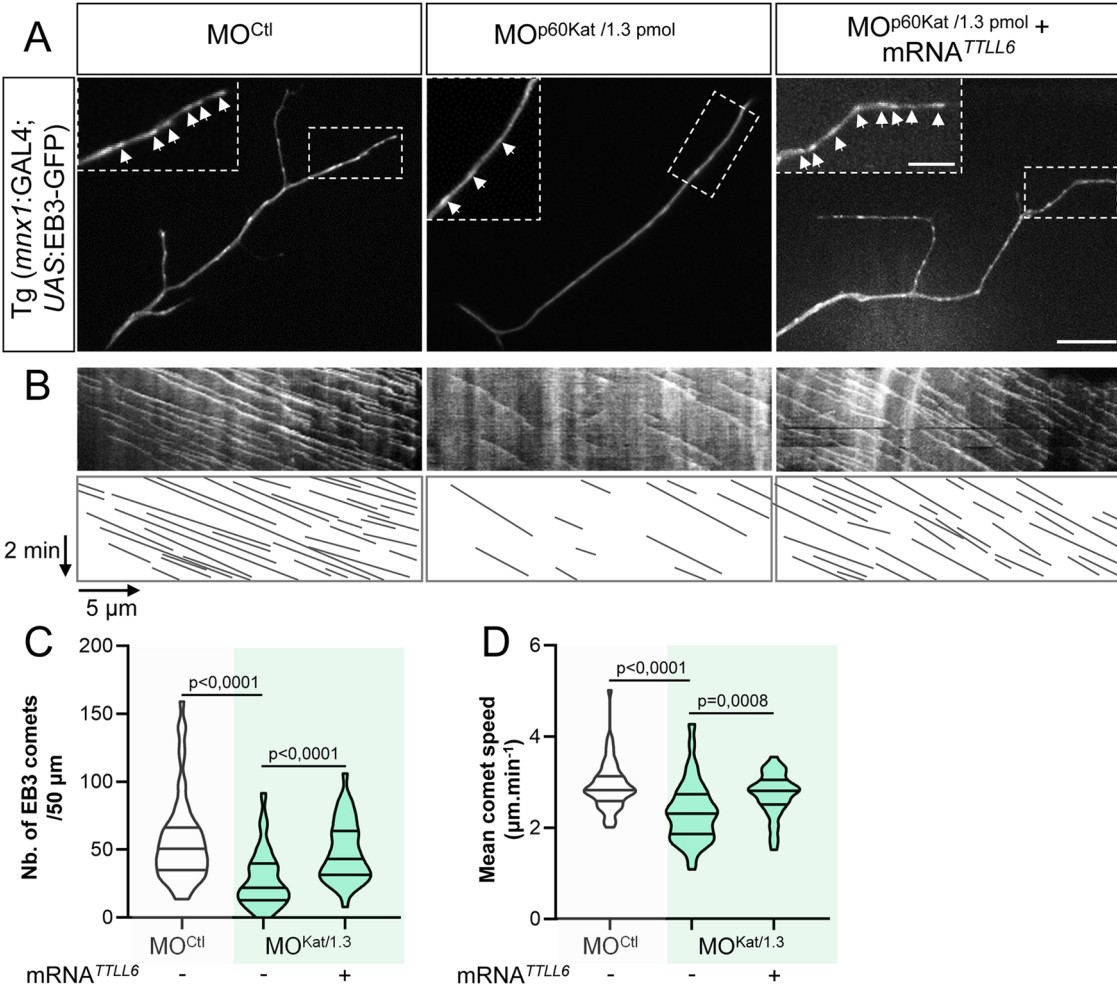

**Figure 7. TTLL6 overexpression rescues the defects of MT dynamics caused by the partial loss of p60-Katanin MT-severing activity in navigating motor axons.**

(A) Representative z-projection still images of EB3-GFP comet time-lapse recordings in sMN axons of 52-hpf Tg(*Mnx1*:GAL4;*UAS*:EB3-GFP) larvae injected with MO$^{Ctl}$, MO$^{p60Kat/1.3pmol}$, or co-injected with MO$^{p60Kat/1.3pmol}$ and mouse *TTLL6* mRNAs (related to Movie EV4). Arrowheads point at MT plus ends (i.e., EB3-GFP comets). Insets are higher magnifications of the boxed axonal portion. Scale bars: 5 μm and 2 μm (insets). (B) Upper panels: Representative kymograms of a 5-min EB3-GFP recording. Lower panels: Schematic representation of the corresponding kymograms illustrating the EB3-GFP traces. Horizontal bar, 5 μm; vertical bar, 2 min. (C, D) Mean number of EB3 comets per 50 μm of axon (C) and mean EB3 comet velocity (D). MT plus-end dynamics was monitored and quantified in a non-blind manner in 58, 74 and 38 dorsally projecting sMN axons of 72-hpf larvae injected with MO$^{Ctl}$, MO$^{p60Kat/1.3pmol}$ or co-injected with MO$^{p60Kat/1.3pmol}$ and mouse *TTLL6* mRNAs, respectively. These axons were obtained from at least 6 larvae per condition and pooled from four independent experiments. Violin Plots; horizontal bars indicate the median ± the 1st and 3rd quartiles. Kruskal–Wallis ANOVA test with Dunn's post hoc test. *P* values are displayed on graphs. Source data are available online for this figure.

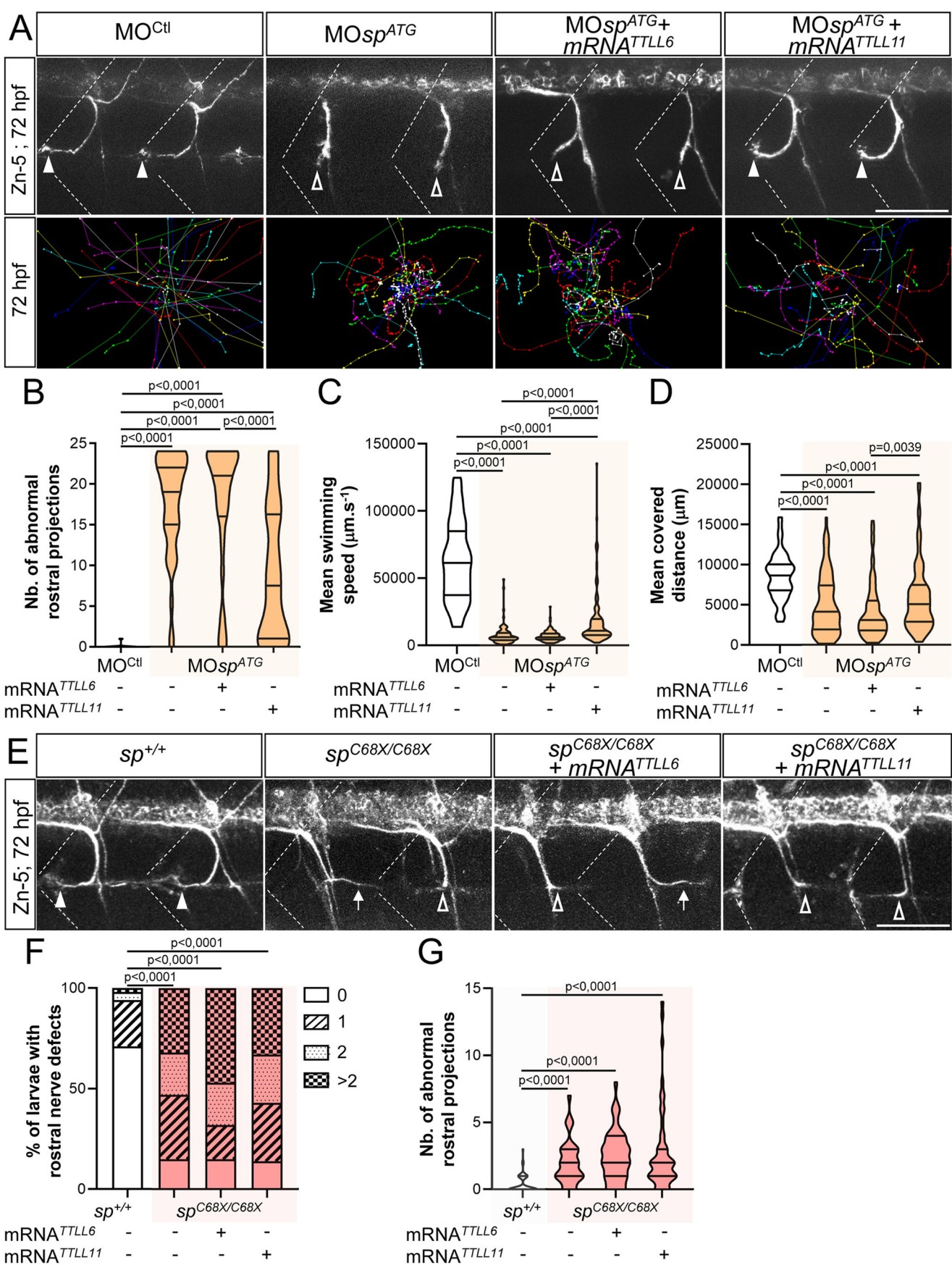

◀

**Figure 8. Selective regulation of spastin activity by TTLLL11 controls zebrafish motor circuit wiring and larval motility.**

(A) Upper panels: Immunolabelling of sMN axon tracts using Zn-5 antibodies in 72-hpf Tg(*Hb9*:GFP) larvae injected with MO$^{Ctl}$ ($n = 69$), MO*sp*$^{ATG}$ ($n = 79$) or co-injected with MO*sp*$^{ATG}$ and mouse *TTLL6* ($n = 63$) or *TTLL11* ($n = 78$) mRNAs. Dotted lines delineate lateral myosepta. Lower panels: Tracking analysis of 72-hpf larvae injected with MO$^{Ctl}$ ($n = 71$), MO*sp*$^{ATG}$ ($n = 90$) or co-injected with MO*sp*$^{ATG}$ and mouse *TTLL6* ($n = 98$) or *TTLL11* ($n = 124$) mRNAs in a touch-escape response test. Each line represents the trajectory of one larva after touch stimulation while the distance between two dots indicates the distance covered by a larva between two consecutive frames. Scale bar: 5 mm. (B–D) Quantifications of the sMN (B) and locomotor defects (C, D) of larvae analysed in (A) and pooled from three independent experiments. (B) Mean number of abnormal rostral nerves. (C, D) Mean swimming speed (C) and covered distance (D). (E) Immunolabelling of sMN axons in 72-hpf $sp^{+/+}$ ($n = 52$), $sp^{C68X/C68X}$ ($n = 34$) and $sp^{C68X/C68X}$ larvae injected with mouse *TTLL6* ($n = 47$) or *TTLL11* ($n = 49$) transcripts using Zn-5 antibody. (F, G) Quantifications of rostral nerve defects in larvae analysed in (E) and pooled from three independent experiments. (F) Percentage of larvae with rostral nerve defects. (G) Mean number of abnormal rostral nerves (i.e., defasciculated or missing) per larva. (A, E) Full and empty arrowheads, respectively, point at normal and defasciculated/missing rostral nerves. Full arrows in (E) indicate caudally oriented "rostral" nerves. Scale bar: 50 μm. (B, F, G) Non-blind quantifications were performed on 24 spinal hemisegments located around the yolk tube per larva. (B–D, G) Violin Plots; horizontal bars indicate the median ± the 1st and 3rd quartiles. Kruskal–Wallis ANOVA test with Dunn's post hoc test. (F) Chi² test. *P* values are displayed on graphs. Source data are available online for this figure.

MO$^{katna1aug1}$: 5′-CATCCTGTAAGTTAAAGTGGTCAGT-3′ (Butler et al, 2010),

MO$^{TTLL6}$: 5′-CTGGTGTCCCCATTCTGATCTCTTC-3′ (Pathak et al, 2011), and

MO$^{TTLL11}$ : 5′-CGGCTGATTTGTTATCTCATCTAGG-3′,

MO*sp*$^{ATG}$: 5′-GCTGAAACAGCCACCGAAGAAGCC-3′ (Jardin et al, 2018).

MO$^{p60Kat}$ was injected at 1.3 and 3.4 pmol/embryo, MO$^{katna1aug1}$ at 0.9 pmol/embryo, MO$^{TTLL6}$ at 0.2 pmol/embryo, MO$^{TTLL11}$ at 0.8 pmol/embryo and MO*sp*$^{ATG}$ at 0.2 pmol/embryo. Universal control MO$^{ctl}$ morpholino was injected at all these doses depending on the controlled knockdown experiment. All morpholinos were injected at the two-cell stage into the yolk just below the cells. Human full-coding *KATNA1* cDNA was isolated from a human foetal brain Marathon-ready cDNA collection (Clontech, Invitrogen) with the following forward and reverse primers: 5′-ATA TAGGATCCATGTACCCATACGATGTTCCAGATTACGCT AG TCTTCTTATGATTAGTGAG-3′ and 5′-ATATATCTAGATTAG-CATGATCCAAACT CAAATATC-3′, and subsequently cloned into pCS2+ BamH1/XbaI restriction sites for rescue experiments. Zebrafish *spastin* full-length cDNA was amplified and HA-tagged by PCR from the IMAGE clone BG728071 with 5′-ATACTCGAG-CAAGCTTGATTTAGGTGA-3′ forward and 5′-GGCTCTAGAT-CAAGCGTAGTCAGGCACGTCGTAAGGGTAACTAGCGCCTA CGCCAGTCGTGTCTCCGT-3′ reverse primers, and cloned into pCS2+ using the XhoI/XbaI restriction sites included in the primers. Human *KATNA1*, mouse *TTLL6* and *TTLL11* (cDNAs provided by C Janke), as well as zebrafish *spastin* mRNAs were in vitro-transcribed from the corresponding linearised pCS2+ constructs using the SP6 mMessage mMachine kit (Ambion) and injected at the one-cell stage. For rescue experiments, *KATNA1* mRNA was injected at 120 or 180 pg/embryo, *spastin* transcript at 200 pg/embryo, and mouse wild-type or catalytically dead *TTLL6* or *TTLL11* mRNAs at 120 pg/embryo.

## Genomic DNA isolation and genotyping

Genomic DNA was isolated by incubating larval heads 2 h at 55 °C in lysis buffer (100 mM Tris Hcl/2 mM EDTA pH8/0.2% Triton X-100/250 μg/mL proteinase K). Homozygous and heterozygous *katna1* mutants were identified by PCR amplification followed by DNA sequencing (GENEWIZ). The primers used for genotyping were the following:

Katna1sa18250_FOR: 5′-GTAGTACGGAAATCCTCTGTCC-3′

Katna1sa18250_REV: 5′-TTGCTTTGATCTAAGAAACCGG-3′.

*Spastin* ($sp^{C68X/C68X}$) mutants were genotyped as previously described in Jardin et al (2018).

## RNA extraction and RT-qPCR analysis

For sequence analysis of *katna1* mRNA, the total mRNA of 24-hpf *katna1*$^{+/+}$ and maternal zygotic *katna1*$^{-/-MZ}$ zebrafish embryos were extracted with Trizol according to the manufacturer's instructions. *Katna1* transcripts were reverse-transcribed and PCR-amplified using the SuperScript™ III One-Step RT-PCR System with Platinum® Taq (ThermoFisher Scientific, France) from 200 ng of *katna1*$^{+/+}$ and *katna1*$^{-/-MZ}$ RNA extracts using the following primers:

Katna1RTPCRex3-4_FOR: 5′-ATGTGGAGCACAGATCGTCT CCATGTG-3′

Katna1RTPCRex5-6_REV: 5′-CAGCAATGTCATCCCATGTG ACATTGGG-3′.

In RT-qPCR analysis of *katna1* homologous gene expression levels, RNAs were extracted from 5 independent pools of 10 *katna1*$^{+/+}$, 10 *katna1*$^{-/-MZ}$ and one pool of 10 wild-type embryos (at 24 hpf) using the RNeasy Mini-kit (Qiagen, France) and reverse-transcribed using the Superscript RT II Kit with random hexamers (Invitrogen, ThermoFisher Scientific, France). qPCR was performed using a SYBR Green master mix (EurobioGreen qPCR Mix, Hi-ROX; Eurobio Scientific, France) using the primers listed in Table 1. The relative quantification method was used to calculate the expression levels of the genes of interest normalised to *lsmb12* and relative to the cDNAs from 24-hpf wild-type embryos. Values are shown as means ± SEM.

## Whole-mount immunohistochemistry

Zebrafish embryos were fixed in PBS/4% paraformaldehyde during 2 h at room temperature, washed three times with PBT1% (1% Triton X-100 in PBS) and permeabilised using a 0.25% trypsin solution (Gibco) at 25 °C when embryos were older than 24 hpf. Embryos were then blocked for two hours in PBT1% supplemented with 10% normal goat serum and incubated with primary antibodies overnight at 4 °C in PBT1%/1% normal goat serum. The following primary antibodies were used: the anti-synaptotagmin 2 antibody (Znp-1, 1/100; ZIRC, University of Oregon) labelling both pMNs and sMNs, the Zn-5 antibody (Zn-5, 1/150; ZIRC, University of Oregon) recognising the DM-GRASP/

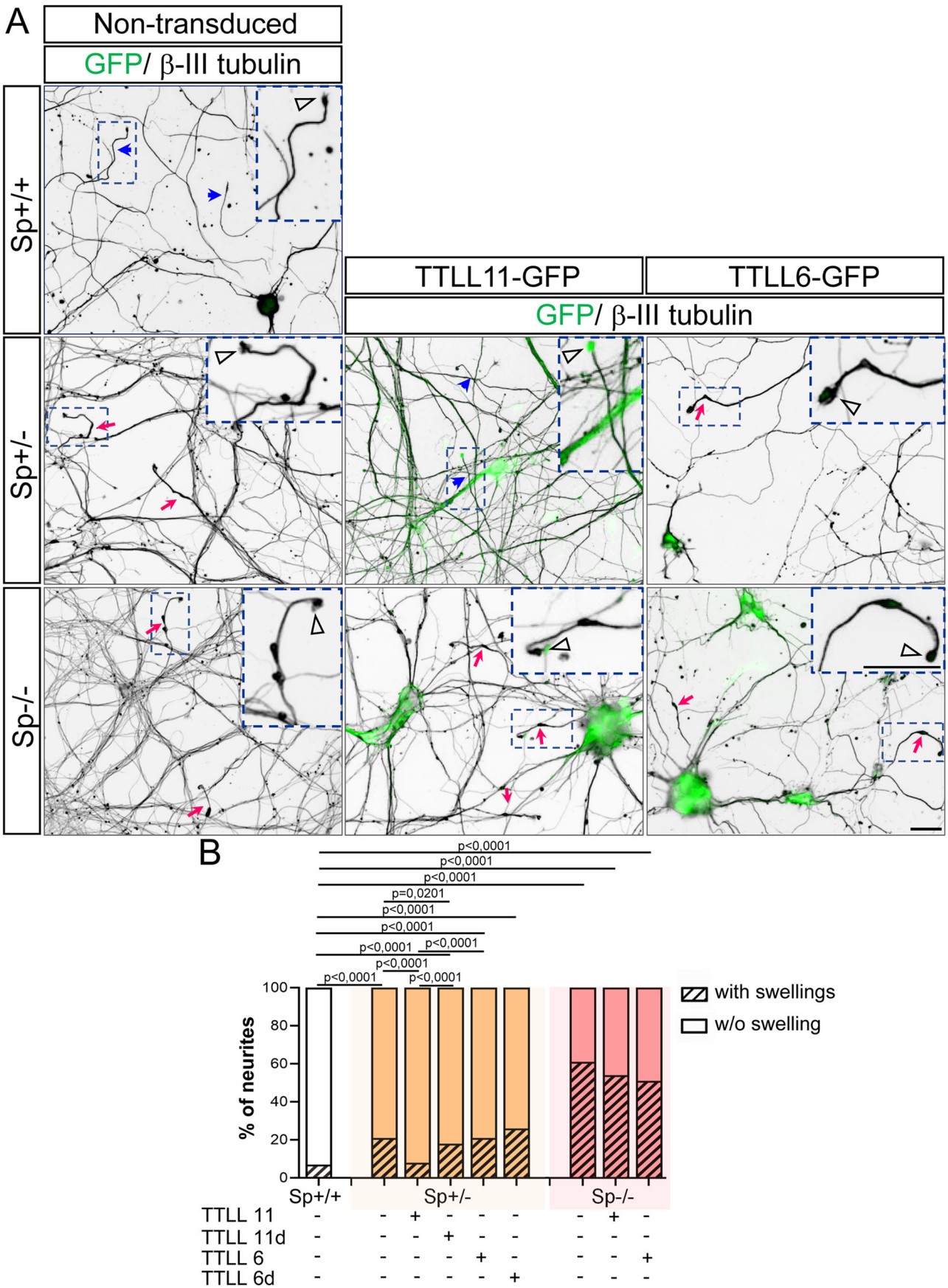

**Figure 9. TTLL11, but not TTLL6, rescues the axonal swelling phenotype caused by spastin haploinsufficiency in mammalian cortical neurons.**

(A) Immunolabelling of β-III tubulin and GFP on DIV9 Sp + /+, Sp + /- and Sp-/- cortical neurons transduced or not at DIV2 with lentiviruses encoding GFP-tagged wild-type or catalytically dead variants (TTLL11d and TTLL6d) of mouse TTLL11 or TTLL6. Blue arrowheads and pink arrows, respectively, indicate non-swollen and swollen axons. Insets are higher magnifications of the distal part of the axons framed in the boxed region of each corresponding image. Arrowheads indicate growth cones. Scale bars: 20 μm. (B) Percentage of swollen axons. Blind quantifications were performed on at least 400 axons per condition pooled from two or three independent experiments. The "**d**" stands for catalytically dead variants of the TTLL enzymes. Chi2 test. *P* values are displayed on graphs. Source data are available online for this figure.

Neurolin transmembrane protein and specifically labelling sMNs, and the rabbit GFP antibody (A11122, 1/1000; Molecular Probes). After several washes in PBT1%, embryos were incubated overnight at 4 °C with the appropriate secondary antibody (Alexa Fluor 488 or 555 at 1/1000, Molecular Probes).

For polyglutamylated tubulin staining, embryos were fixed in Dent's fixative (80% Methanol/20% DMSO) overnight at 4 °C. After rehydration in regressive methanol/PBS-0.5% Tween 20®, embryos were blocked in PBT1% supplemented with 1% DMSO, 1% bovine serum albumin and 5% normal goat serum, and sequentially incubated with primary antibodies, GT335 (1:1000, provided by C Janke) and PolyE (1:1000; provided by C. Janke), and adequate secondary antibodies. Images were acquired using a fluorescence microscope equipped with an Apotome module (Zeiss, Axiovert 200 M), a ×20 objective (NA 0.5), the AxioCam MRm camera (Zeiss) and the Axiovision software (Zeiss). Images were processed with the NIH Fiji software. Each figure panel corresponds to a projection image from a z-stack of 2-μm sections.

For polyE signal quantifications, pMN axons of 26-hpf MO^Ctl-, MO^TTLL6- and MO^TTLL11-injected embryos immunolabelled with polyE antibodies were imaged using a Nikon Eclipse Ti2 microscope equipped with a Nikon AX/AX R confocal module and a ×20 objective (N.A: 0.8) using the NIS-Elements Imaging Software (Nikon Instruments Inc., Melville, USA). Identical settings (exposure time, z-stack thickness, etc.) were used for both control and morphant embryos. Quantifications were performed on maximum z-projection images using the Fiji software. Regions of interest (ROI) were traced along 12 pMN axons (located around the yolk tube) per embryo and the mean PolyE fluorescence intensity was quantified for each axon. The same ROI set was used to measure the mean fluorescence intensity background in the region adjacent to each axon. Each background value was subtracted from the corresponding polyE value before statistical analysis.

### Time-lapse videomicroscopy assays

For in vivo monitoring of spinal motor axon outgrowth, control and MO^p60Kat-injected Tg(*Hb9*:GFP) embryos were anesthetised with E3 medium containing tricaine and embedded in 0.8% low-melting agarose in a 35-mm glass dish (Iwaki). Time-lapse videomicroscopy recording of spinal motor axon outgrowth was carried out at 28 °C in E3 medium (supplemented with tricaine) using a Leica DMI 6000B inverted spinning-disk microscope with a ×40 immersion objective (NA 1.4) and a 491-nm 100-mW Cobolt calypso laser over 30 h. Embryos immobilised at 22 hpf were filmed with a ×20 immersion objective over 48 h. In both cases, z-stacks of 80-μm-thick sections were acquired every 8 min with a step size of 1 μm using an EMCCD camera (Photometrics Quantem 512 SC)

and the Metamorph software (Molecular Devices) and compiled into time-lapse movies.

For time-lapse recording of MT plus-end dynamics, Tg(*Mnx1*:Gal4) larvae were first injected at the one-cell stage with 25 ng of a Tol1-14*UAS*:EB3-GFP plasmid, 25 ng of *Tol1* mRNA and 120 ng of *TTLL6* mRNA (for rescue experiments only) and reinjected at the two-cell stage with control (MO^Ctl) or *p60-katanin* morpholino (MO^p60Kat/1.3pmol). At 52 hpf, larvae were anaesthetised with tricaine and embedded in 1% low-melting point agarose in a petri dish. Time-lapse recordings of EB3-GFP comets were performed at 28 °C in E3 medium supplemented with tricaine using an upright DM6000 Leica microscope (Leica Microsystems SAS, Nanterre, France) equipped with a spinning-disk confocal CSU-X1 (Yokogawa, France) and a long-working-distance ×63 water-dipping objective (NA: 1.00; Zeiss). Z-stacks of 10 planes interspaced of 0.8 μm were acquired every 4 sec over a 5-min period using a Zyla sCMOS camera (Andor, Oxford Instruments) controlled by the MetaMorph software (Molecular Devices) and were compiled into time-lapse videos. EB3 comet density and velocity were estimated on single-plane maximum z-projection videos by using kymograph analysis (Fiji software) with lines (ROI) drawn along dorsally projecting sMN axons as carried out in Fassier et al (2018).

### Touch-evoked escape response test and manual tracking

To assess the motor behaviour of control, mutant, morphant and rescued 72-hpf larvae, we performed a touch-escape response test by applying a tactile stimulus with a pair of forceps and analysing the larval escape behaviour under a Leica M165C binocular stereomicroscope equipped with a Leica IC80 HD camera. Swimming speed and covered distance of each larva were quantified using the Manual Tracking plugin included in the Fiji software as reported in Fassier et al (2010).

### Whole-mount in situ hybridisation

A 3'-fragment of *katna1* cDNA was isolated from a collection of 24-hpf zebrafish embryo mRNAs using the SuperScript® III One-Step RT-PCR system (Invitrogen) with the following forward and reverse primers: 5'-AGAGTGGATTTACTCAAGATCAACC-3' and 5'-AAGCTTGACTTGTACGCAGTGAACC-3'. The RT-PCR product was cloned into the TOPO® TA cloning pcr4 vector (Invitrogen) and sequenced. The *katna1* digoxigenin-labelled sense and antisense riboprobes were synthesised from the linearised recombinant TOPO® TA cloning vector using T7 and T3 RNA Polymerase (Promega) according to the supplier's instructions. Whole-mount in situ hybridisation experiments were performed at 18 and 24 hpf using standard procedures (Macdonald et al, 1994).

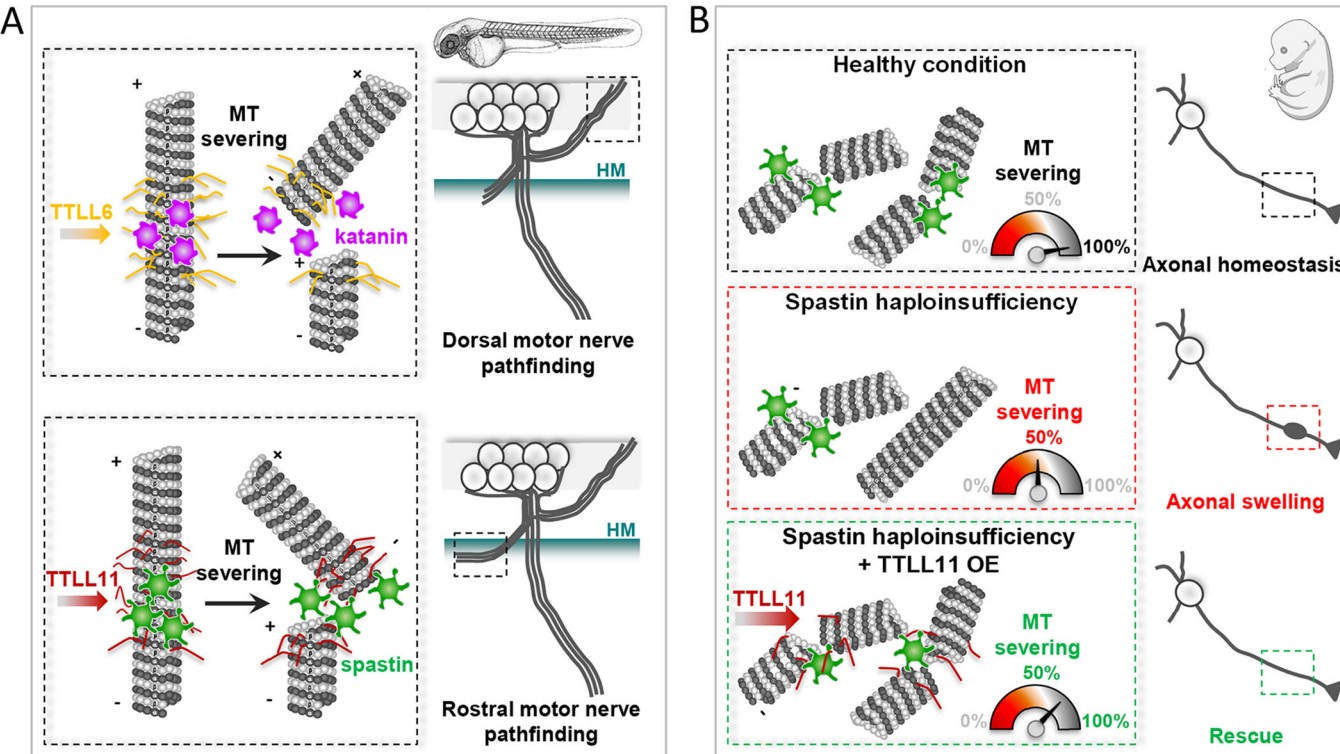

**Figure 10. Schematic representation of MT-severing enzyme selective regulation by specific TTLL enzymes in zebrafish and mammalian neurons.**

(**A**) Upper panel: TTLL6-mediated tubulin polyglutamylation selectively tunes p60-Katanin activity in zebrafish dorsally projecting secondary motor axons to control their targeting. Lower panel: TTLL11-driven MT polyglutamylation is required for accurate axon pathfinding of rostrally projecting secondary motor nerves in zebrafish larvae. (**B**) Spastin haploinsufficiency induces axonal swellings in mammalian cortical neurons (middle panel), which can be selectively rescued by promoting TTLL11-mediated tubulin polyglutamylation (lower panel), most likely through the boosting of residual spastin activity upon the critical threshold (i.e., 50%).

Pictures were acquired with a binocular stereomicroscope (Leica M165C) combined with an HD camera (Leica IC80 HD), adjusted for brightness and contrast with the NIH Image J software.

*In toto* hybridisation chain reaction (HCR) experiments were performed on 72-hpf larvae using *spastin* (NM_212915) and *KATNA1* (NM_001020604) probes designed and synthesised by Molecular Instruments (Molecular Instruments Inc, Los Angeles, USA) following the manufacturer's instructions (Choi et al, 2018). Briefly, embryos were fixed in 4% PFA overnight at 4 °C, washed in PBS, dehydrated and permeabilized through two 10-min washes in 100% MeOH and stored for at least 24 h in fresh MeOH at −20 °C. Embryos were then rehydrated through a graded series of MeOH(75%, 50%, 25%, 0%)/PBS-Tween 20 (PBST) solution baths, permeabilised with proteinase K (10 μg/mL) for 70 min (72 hpf), washed twice in PBS/Tween, post-fixed in 4% PFA for 20 min, and washed several times in PBT1%. After a pre-incubation step in the hybridisation buffer (Molecular Instruments), embryos were incubated overnight with the specific probes diluted in the hybridisation buffer. The following day, embryos were sequentially washed several times with the washing buffer and the sodium chloride sodium citrate-Tween 20 (SSCT) solution (Molecular Instruments). Hairpins were heated to 95 °C for 90 sec and cooled for 30 min before being incubated with embryos overnight, and subsequently washed several times with SSCT. Embryos were flat-mounted in a 90% glycerol solution and imaged using a Nikon

Eclipse Ti2 microscope equipped with a Nikon AX/AX R confocal module and a 40x oil-immersion objective (N.A: 1.3) using the NIS-Elements Imaging Software (Nikon Instruments Inc., Melville, USA). Images were processed with the NIH Image J software. Each figure panel corresponds to a projection image from a z-stack of 0.5-μm sections.

## Primary cultures of cortical neurons, lentiviral infection and immunolabelling

Primary cultures of cortical neurons were prepared as described in Tarrade et al (2006) from Sp + / +, Sp + /- and Sp-/- mice (provided by the Kneussel lab; Lopes et al, 2020) at E14.5 days post-coitum. Cortical neurons were plated at 600,000 neurons per 35-mm Petri dishes and maintained for at least 9 days at 37 °C in 5% $CO_2$. One-third of the medium was changed every three days. Cortical neurons were fixed at 6, 7 and 9 days with 4%PFA in 4% sucrose/PBS for 15 min at 37 °C, permeabilised with 0.1% Triton-100 in PBS for 5 min, blocked in 3% BSA/ 5% normal goat serum in PBS for 1 h and incubated overnight at 4 °C with primary antibodies (β-III tubulin/anti-tuj-1, 1/1000, Biolegend no. 801202 and anti-GFP, 1/1000, Torrey Pines Biolabs no. TP-401) diluted in the blocking solution. After several washes in PBS, neurons were incubated with secondary antibodies (anti-mouse-568, 1/1000, Molecular probes A11019; anti-rabbit-488, 1/1000, Molecular

**Table 1. Primers used for RT-qPCR analysis.**

| Gene | Forward sequence (5'–3') | Reverse sequence (5'–3') |
|---|---|---|
| *spast* | AAGGCAGACAGAGCCAGAAAA | TGAAAGCAGATTGCCCAGAAG |
| *katnal2* | GAGGCTGCCAAGCGATTAGT | GGCGACAGGATTCCAGTGAA |
| *fign* | CCCGCACAAGCATTCATCAG | CCACTGCATCTTTAAGCCTGT |
| *fignl1* | TCCAGGCACTGGTAAAACCC | CCTTCCCCGACCCATTTTGA |
| *katnal1* | GTCTGCAGAGATGCGTCGAT | ATGGTGACCGGCATCTGAAG |
| *vps4a* | AGCATCTCATGGGTGCGATT | GAGCTTCTTTGGCTCCCTCC |
| *lsm12b* | GAGACTCCTCCTCCTCTAGCAT | GATTGCATAGGCTTGGGACAAC |

probes A11008), sir-actin (1/2000, Cytoskeleton, no. CY-SC001) and DAPI for 1 h at RT. Neurons were then rinsed in PBS, mounted using the ProLong Gold mounting medium (Invitrogen no. P36934) and imaged using an epifluorescent microscope (DM6000, Leica) with a ×40 objective (N.A: 0.5). The number of neurite swellings per 100 nuclei was determined for each condition. For rescue experiments, Sp + /- and Sp-/- cortical neurons were treated at DIV2 (2 days in vitro) with lentivirus suspensions encoding WT (active) or catalytically dead mouse TTLL6 or human TTLL11. Viruses were produced as described previously (Bodakuntla et al, 2020). Primary cortical neurons were transduced with a volume of viral suspension able to transduce more than 97% of neurons. This volume was determined for each lentiviral construct by infecting the primary neurons with serial dilutions of the supernatant containing the viral particles. Cells were fixed at DIV9 and immunolabelled as described above for the non-transduced cells. To ease the identification of axonal swellings in the dense neural network at DIV9, the number of swollen axons was estimated in at least 100 field of view acquired at the ×40 objective (N.A: 0.75) and related to the total number of analysed axons. Each set of experiments was reproduced three times independently and blindly analysed. Animal care and use were performed in accordance with the recommendations of the European Community (2010/63/UE) for the care and use of laboratory animals. Experimental procedures were specifically approved by the Ethics committee of the Institut Curie CEEA-IC #118 (authorisation #37315-2022051117455434 v2 given by National Authority) in compliance with the international guidelines.

## Statistical analysis

All data were obtained from at least two to three independent experiments. Statistical significance of the data was evaluated using the non-parametric Mann–Whitney test when comparing two groups assuming non-Gaussian distributions. The Kruskal–Wallis ANOVA test with Dunn's post test and the one-way ANOVA test with Bonferroni's post test were used when comparing more than two groups assuming non-Gaussian or Gaussian distribution, respectively. The *Chi*² test was used to compare the distribution of the zebrafish motor neuron defects in different experimental conditions. Data distribution was tested for normality using the D'Agostino and Pearson normality test. All Statistical analyses were performed using GraphPad Prism 9.00 (GraphPad Software, San Diego, CA).

## Data availability

This study includes no data deposited in external repositories.

The source data of this paper are collected in the following database record: biostudies:S-SCDT-10_1038-S44318-024-00307-x.

## Peer review information

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

## Acknowledgements

This work was also supported by the DIM C-BRAINS, funded by the Conseil Régional d'Ile-de-France. This work was supported by research grants to (i) JH from the Association Française contre les Myopathies (AFM), the Emergence Programme from the UPMC (University Paris 6) and the Association Strümpell-Lorrain-HSP France, (ii) CF from the Association Strümpell-Lorrain-HSP France (AO 2015) and the French National Research Agency (ANR) (ANR-20-CE16-0019). CF and MM were jointly supported by grants from the Association Strümpell-Lorrain-HSP France (AO 2019), the Tom Wahlig Foundation (2019) and the AFM (23695). MMM was supported by the France Alzheimer grant ALZcode. CJ was supported by the ANR awards ANR-17-CE13-0021 and ANR-20-CE13-0011 and the Fondation pour la Recherche Medicale (FRM) grant (MND202003011485). DTM, NJ and NB were recipients of PhD fellowships from the "Ministère de l'Enseignement supérieur, de la Recherche et de l'Innovation (MESRI)" (DTM and NJ) and the European MARIE SKŁODOWSKA-CURIE Doctoral Network EGRET-AAA (NB). Furthermore, the authors warmly thank Susanne Bolte and Richard Schwartzman as well as Stéphane Fouquet from the imaging facilities of the "Institut de Biologie Paris Seine" (IBPS / Sorbonne University) and "Institut de la Vision" (IdV/ Sorbonne University) for their assistance with the spinning-disk microscopy. We are also grateful to Alex Bois, Stéphane Tronche, Aldelkrim Mannioui from the IBPS aquatic facility and Karine Duroure and Nicolas Vareillaud from the IdV aquatic facility for fish management and care. We are grateful to Katia Belloul, Colin Jouhanneau, Virginie Dangles-Marie and Helene Gautier from the Institut Curie animal facility for assistance with mouse breeding, and to Veronique Henriot and Veronique Marsaud from the Institut Curie L3 facility for help with lentivirus production. Finally, the authors would like to thank all the Nicol/ Fassier lab members for their insightful comments throughout the revision process.

## Author contributions

**Daniel Ten Martin**: Formal analysis; Investigation; Visualisation. **Nicolas Jardin**: Formal analysis; Investigation; Visualisation. **Juliette Vougny**: Investigation. **François Giudicell**: Investigation. **Laïla Gasmi**: Investigation. **Naomi Berbée**: Investigation. **Véronique Henriot**: Resources; Investigation. **Laura Lebrun**: Resources; Investigation. **Cécile Haumaître**: Formal analysis; Investigation. **Matthias Kneussel**: Resources. **Xavier Nicol**: Supervision; Writing—review and editing. **Carsten Janke**: Conceptualisation; Resources; Funding acquisition; Writing—review and editing. **Maria M Magiera**: Conceptualisation; Resources; Supervision; Funding acquisition; Investigation; Writing—review and editing. **Jamilé Hazan**: Conceptualisation; Formal analysis; Supervision; Funding acquisition; Validation; Methodology; Writing—original draft; Project administration; Writing—review and editing. **Coralie Fassier**: Conceptualisation; Resources; Formal analysis; Supervision; Funding acquisition; Validation; Investigation; Visualisation; Methodology; Writing—original draft; Project administration; Writing—review and editing.

Source data underlying figure panels in this paper may have individual authorship assigned. Where available, figure panel/source data authorship is listed in the following database record: biostudies:S-SCDT-10_1038-S44318-024-00307-x.

## Disclosure and competing interests statement

The authors declare no competing interests.

# Expanded View Figures

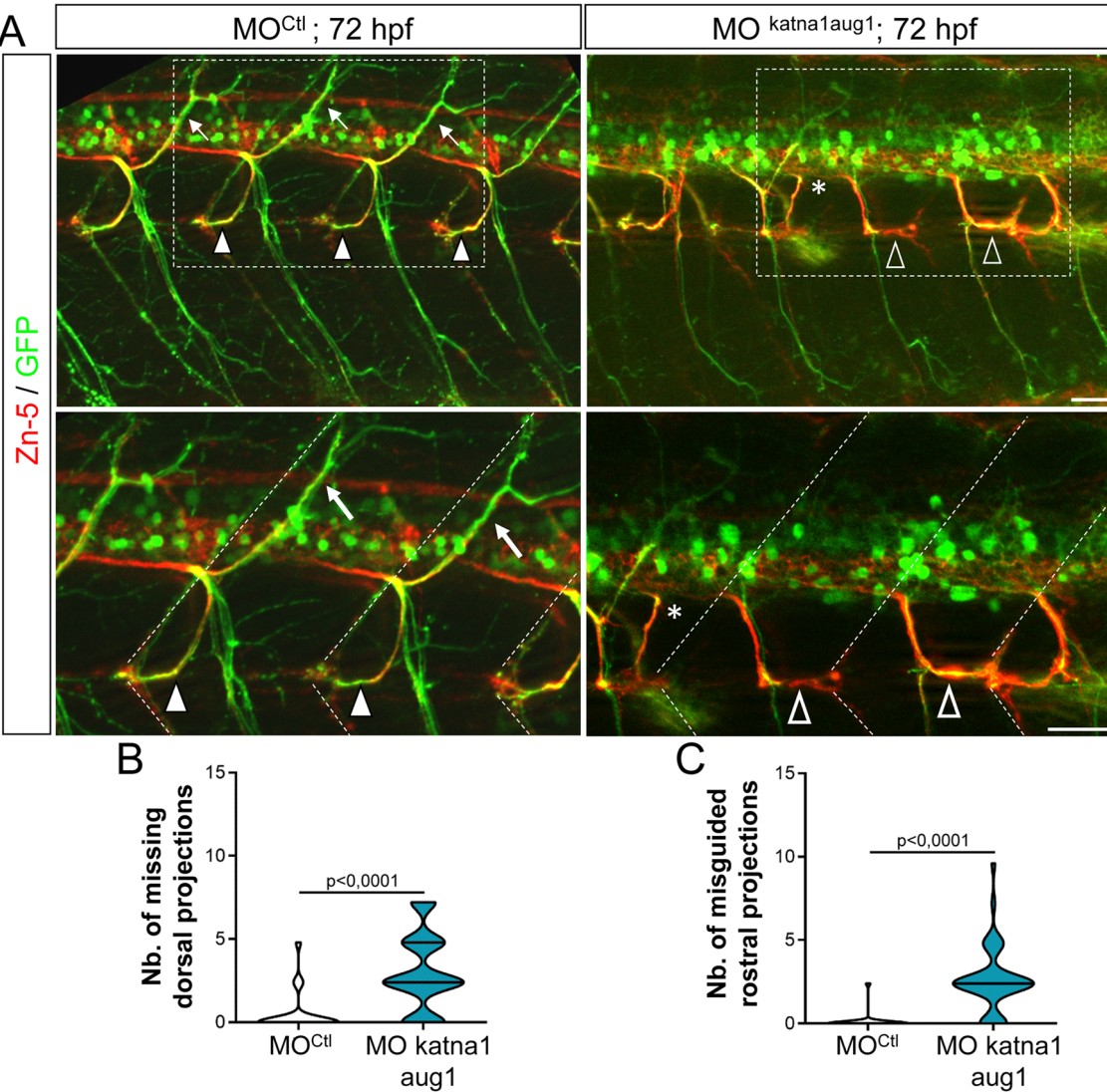

**Figure EV1.  *p60-katanin* knockdown with MO^katna1aug1 morpholino induces similar spinal motor axon defects to MO^p60Kat morpholino.**

(A) Immunolabelling of secondary motoneuron (sMN) axon tracts in 72 h post-fertilisation (hpf) transgenic Tg(*Hb9*:GFP) larvae injected with control ($n = 40$) or MO^katna1aug1 ($n = 40$) morpholinos, using Zn-5 and GFP antibodies. Lateral views of the trunk, anterior to the left. Bottom panels represent higher magnifications of the boxed region in the corresponding top panels. Dotted lines delineate lateral myosepta. Full arrowheads and full arrows point at normal rostral and dorsal nerves, respectively. Empty arrowheads show misguided rostral nerves. Asterisks indicate ectopic sorting points of sMN axons from the spinal cord. (B, C) Quantifications of sMN defects in larvae analysed in panel A and pooled from three independent experiments. Mean number of missing dorsal nerves (B) and misguided rostral nerves (C) per larva. Non-blind quantifications were performed on 24 spinal hemisegments located around the yolk tube per larva. Violin Plots; horizontal bars indicate the median ± the 1st and 3rd quartiles. Mann–Whitney test. *P* values are displayed on graphs. Source data are available online for this figure.

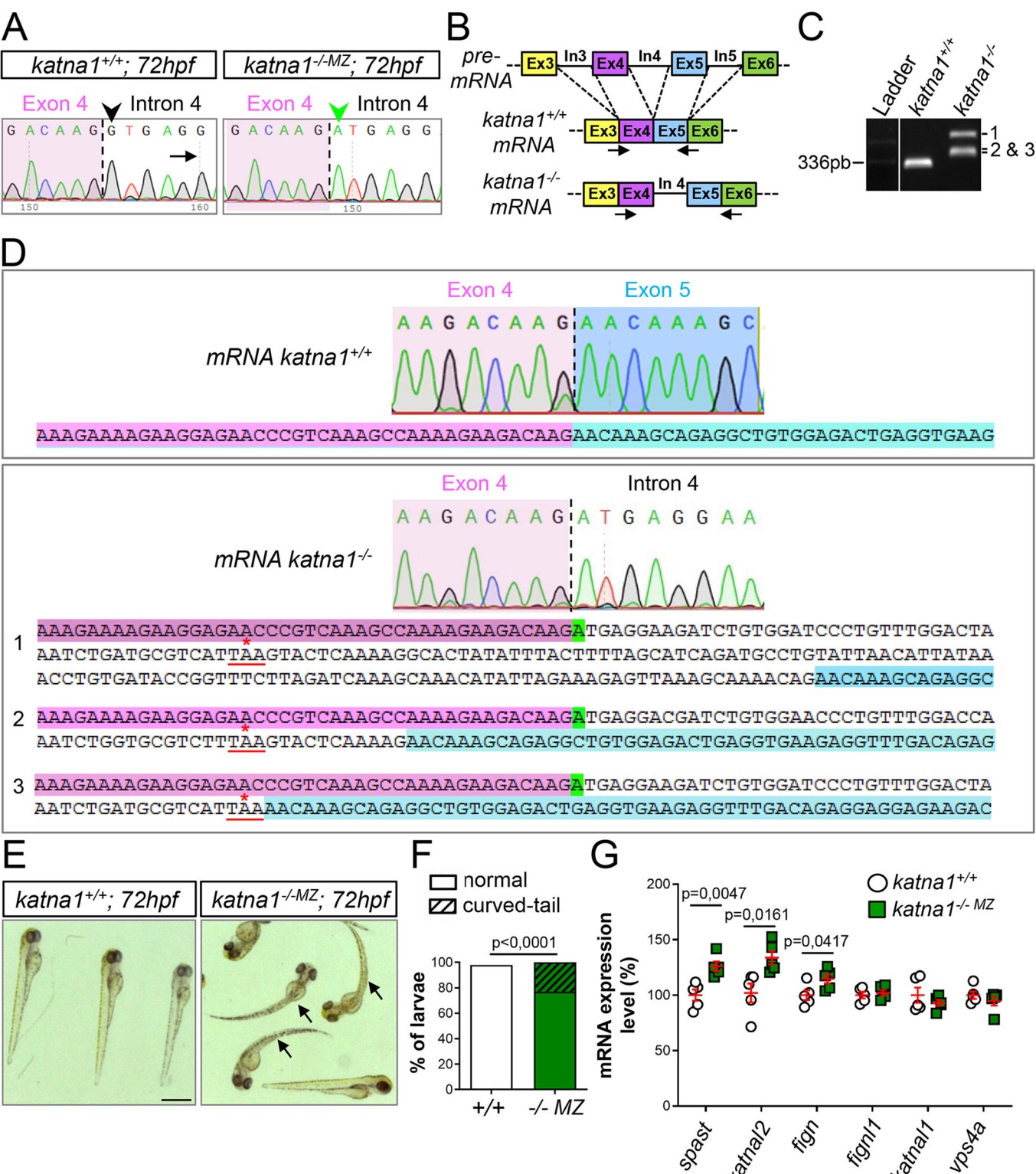

**Figure EV2.  Molecular and morphological characterisation of zebrafish *katna1* mutants.**

(A) Sequence analysis of control and *katna1* mutant genomic DNA. Dotted line indicates the junction between exon 4 (pink background) and intron 4 (white background). The green arrowhead points at the nucleotide substitution (G > A) affecting the donor splice site of intron 4 (G in control, black arrowhead) in *katna1* mutants. (B) Schematic representation of the RT-PCR strategy used to test the impact of the *katna1* splice-site mutation on *katna1* mRNA splicing. Dotted lines indicate intron splicing. Arrows represent the primers used for RT-PCR analysis. Primers were designed on exon/intron junctions to avoid contamination by genomic DNA amplification. In: intron; Ex: exon. (C) RT-PCR analysis of *katna1* intron-4 splicing on total RNA extracts from *katna1*$^{+/+}$ and *katna1*$^{-/-\ MZ}$ maternal zygotic mutant embryos. Homozygous *katna1*$^{-/-\ MZ}$ embryos lack wild-type transcript and show different populations of misspliced transcripts (1, 2 and 3). (D) Sequence analysis of *katna1* misspliced transcripts. Sequences corresponding to exon 4, intron 4 and exon 5 are respectively indicated in pink, white and blue. The splice-site mutation is highlighted in green. Misspliced transcripts include various sized insertions of intron 4, which all lead to a frameshift and the occurrence of a premature stop codon at the same amino-acid position (red asterisk). (E) Gross morphology of 72-hpf control (*katna1*$^{+/+}$) and maternal zygotic *katna1* mutant (*katna1*$^{-/-MZ}$) larvae. Arrows point at the curved-tail phenotype of some mutant larvae. Scale bar: 0.5 mm. (F) Percentage of larvae exhibiting a curved-tail phenotype. Chi$^2$ test. (G) RT-qPCR analysis showing the differential expression levels of *p60-katanin*-related genes from the ATPase Meiotic Clade in control and *katna1*$^{-/-\ MZ}$ larvae. RNAs were extracted from 5 independent pools of 10 *katna1*$^{+/+}$, 10 *katna1*$^{-/-MZ}$ and one pool of 10 wild-type embryos at 24 hpf. Unpaired *t* test. Values are shown as means ± SEM. (F, G) *P* values are displayed on graphs. Source data are available online for this figure.

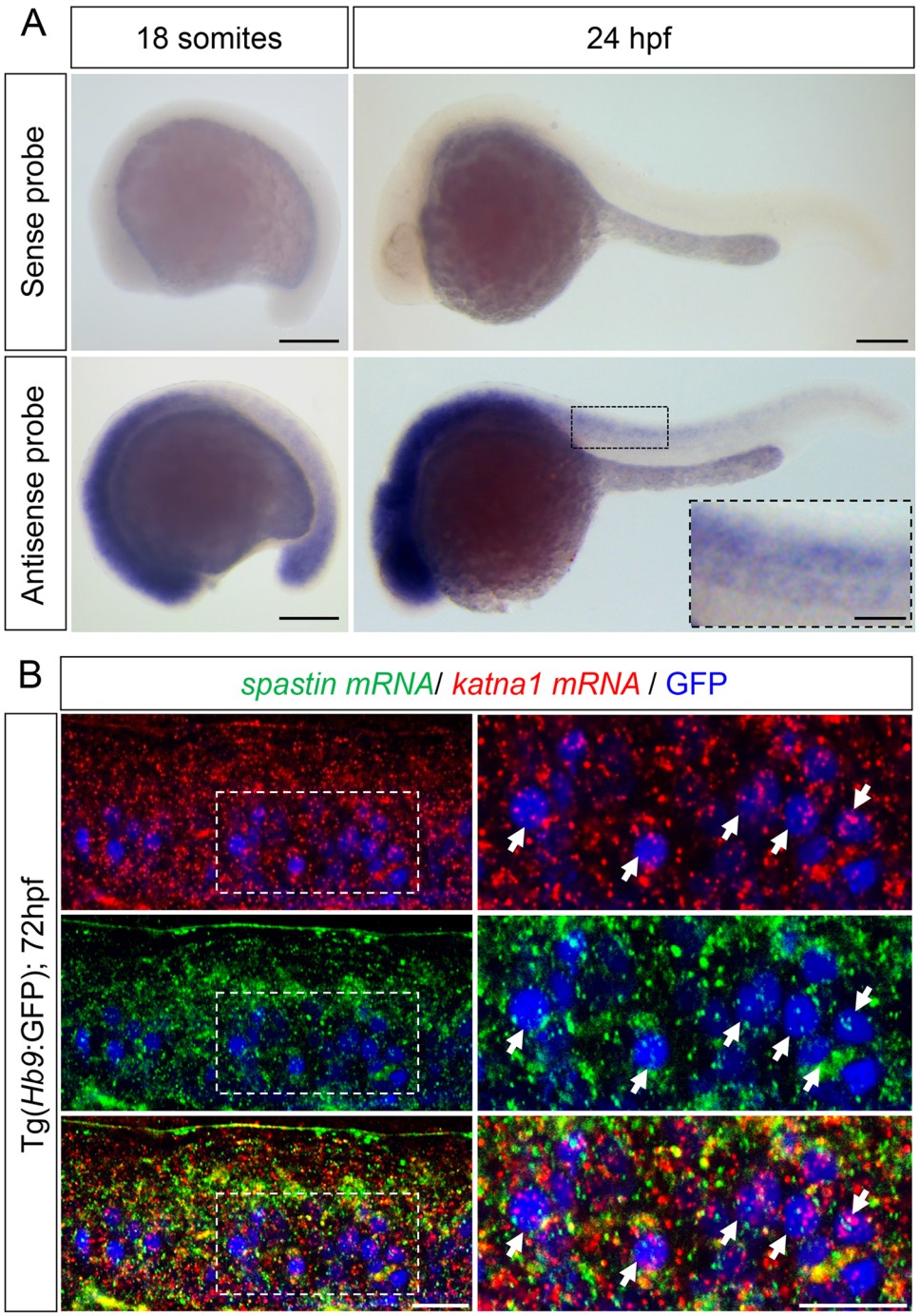

**Figure EV3.** *p60-katanin* and *spastin* transcripts are both expressed in developing spinal motor neurons.

(**A**) Whole-mount in situ hybridisation with *p60-katanin* sense (upper panel) and antisense (bottom panel) riboprobes at 18 somites and 24 h post-fertilisation (hpf). Lateral views of the embryos, anterior to the left. *P60-katanin* is highly enriched in the developing nervous system at both 18 somites and 24 hpf, two stages at which the axons of primary (pMN) and secondary (sMN) motor neurons exit the spinal cord to navigate towards their muscle targets. Scale bars: 200 μm. (**B**) *In toto* hybridisation chain reaction (HCR) on 72-hpf Tg(*Hb9*:GFP) larvae using zebrafish *spastin* and *katna1* probes. Lateral view of the trunk, anterior to the left. Right panels are higher magnifications of boxed region of the corresponding left panel. Arrows point at spinal motor neurons co-expressing *katna1* and *spastin* transcripts. Scale bars: 20 μm. Source data are available online for this figure.

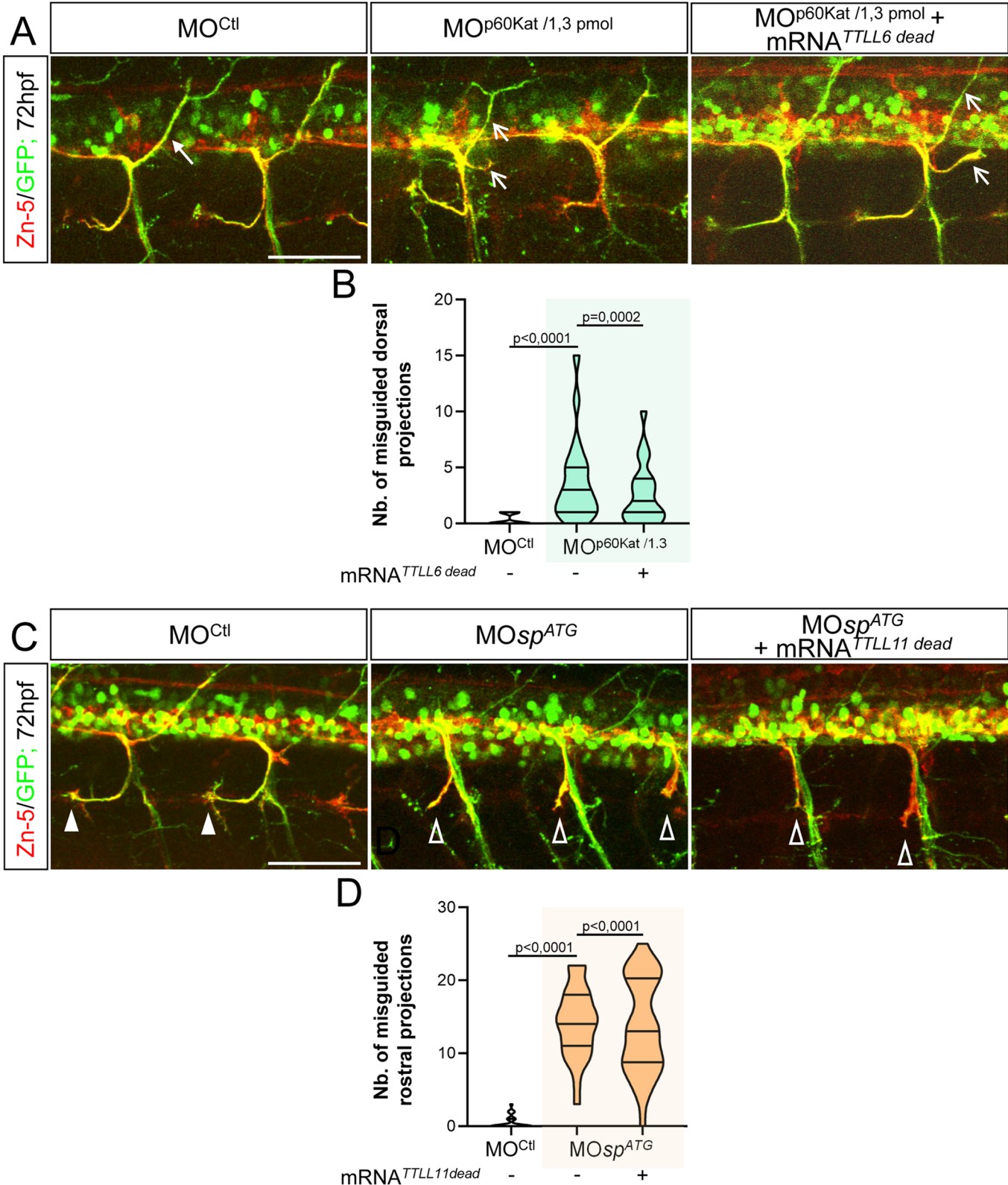

◀

**Figure EV4.  Overexpression of a catalytic-dead variant of TTLL6 or TTLL11 respectively fails to rescue the axon pathfinding errors associated with p60-Katanin or Spastin partial knockdown.**

(A) Immunolabelling of sMN axon tracts using Zn-5 and GFP antibodies in 72-hpf Tg(*Hb9*:GFP) larvae injected with MO$^{Ctl}$ ($n = 29$), MO$^{p60Kat/1.3pmol}$ ($n = 21$) or co-injected with MO$^{p60Kat/1.3pmol}$ and the mRNA encoding a catalytic-dead variant of TTLL6 (TTLL6$^{dead}$) ($n = 23$). Full and empty arrows respectively point at normal and misguided dorsal projections. (B) Mean number of split/misguided dorsal nerves per larva. Quantifications were conducted on the larval set analysed in (A). (C) Immunolabelling of sMN axon tracts using Zn-5 and GFP antibodies in 72-hpf Tg(*Hb9*:GFP) larvae injected with MO$^{Ctl}$ ($n = 27$), MO$sp^{ATG}$ ($n = 27$) or co-injected with MO$sp^{ATG}$ and the transcript encoding a catalytic-dead variant of TTLL11 (TTLL11$^{dead}$) ($n = 34$). Full and empty arrowheads indicate normal and misguided rostral projections, respectively. (D) Mean number of misguided rostral nerves per larva. Quantifications were conducted on the larval set analysed in (B). (A, C) Scale bars: 50 µm. (B, D) Non-blind quantifications were performed on 24 spinal hemisegments located around the yolk tube per larva. Analysed larvae were pooled from three independent experiments. Violin Plots; horizontal bars indicate the median ± the 1st and 3rd quartiles. Kruskal–Wallis ANOVA test with Dunn's post hoc test. *P* values are displayed on graphs. Source data are available online for this figure.

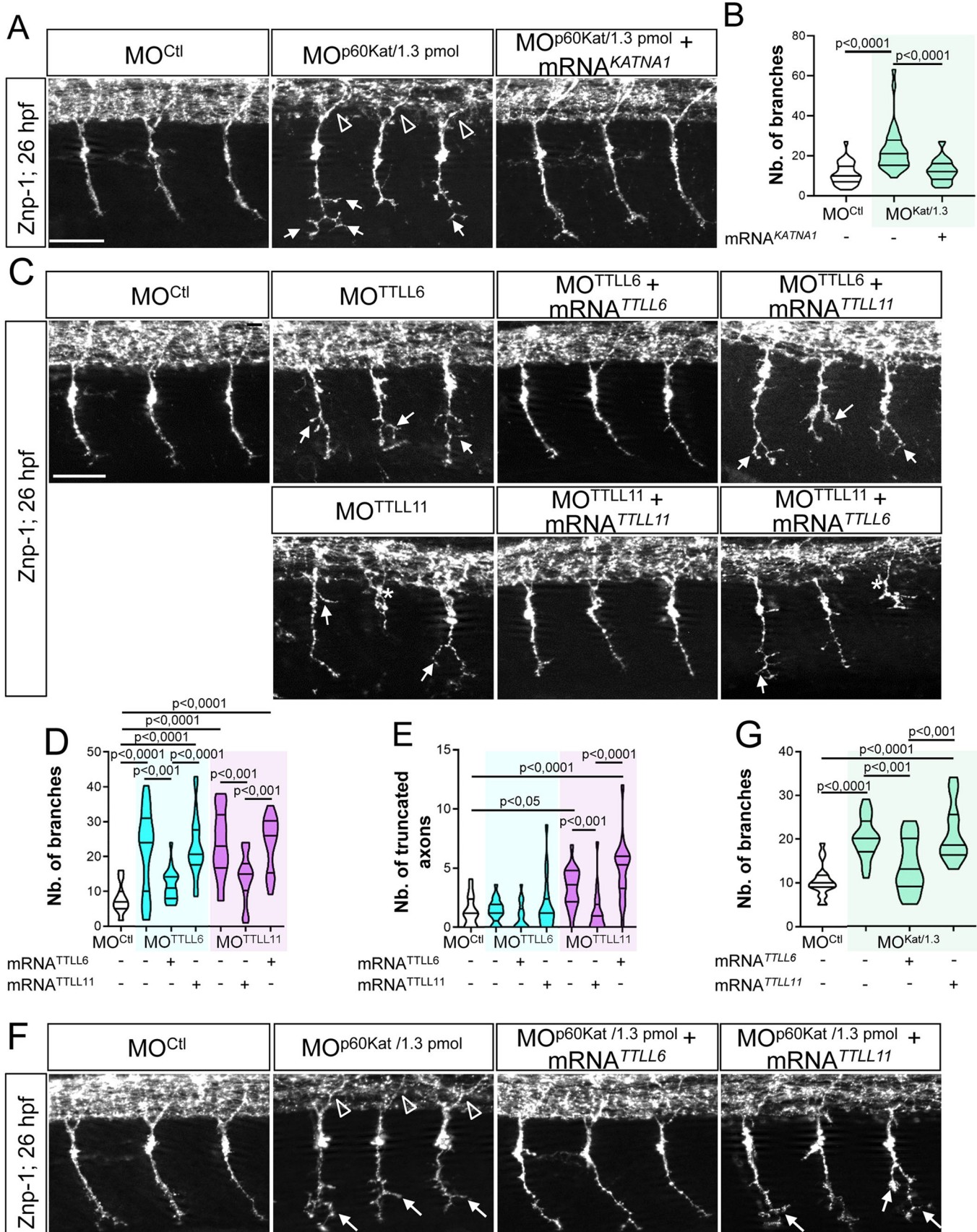

◀  **Figure EV5.   TTLL6 also tunes p60-Katanin-driven pMN axon development.**

(A) Immunolabelling of pMN axons in 26-hpf embryos injected with MO$^{Ctl}$ ($n = 32$), MO$^{p60Kat/1.3pmol}$ ($n = 32$) or co-injected with MO$^{p60Kat/1.3pmol}$ and 120 pg of human *KATNA1* transcripts (MO$^{p60Kat/1.3pmol}$ + mRNA$^{KATNA1}$, $n = 32$) using Znp-1 antibody. (B) Mean number of pMN axon branches per embryo analysed in the panel-A embryo set. (C) Immunodetection of pMN axons with Znp-1 antibody in 26-hpf embryos injected with MO$^{Ctl}$ ($n = 19$), MO$^{TTLL6}$ ($n = 22$), MO$^{TTLL11}$ ($n = 19$) morpholinos or co-injected with MO$^{TTLL6}$ or MO$^{TTLL11}$ and mouse *TTLL6* or *TTLL11* mRNA (MO$^{TTLL6}$ + mRNA$^{TTLL6}$, $n = 20$; MO$^{TTLL6}$ + mRNA$^{TTLL11}$, $n = 21$; MO$^{TTLL11}$ + mRNA$^{TTLL11}$, $n = 19$; MO$^{TTLL11}$ + mRNA$^{TTLL6}$, $n = 18$). (D, E) Mean number of CaP pMN branches (D) and truncated CaP axons per embryo analysed in the panel-C embryo set. (F) Immunostaining of pMN axons with Znp-1 antibody in 26-hpf embryos injected with MO$^{Ctl}$, ($n = 20$), MO$^{p60Kat/1.3pmol}$ ($n = 21$) or co-injected with MO$^{p60Kat/1.3pmol}$ and mouse *TTLL6* (MO$^{p60Kat/1.3pmol}$ + mRNA$^{TTLL6}$, $n = 19$) or *TTLL11* mRNA (MO$^{p60Kat/1.3pmol}$ + mRNA$^{TTLL11}$, $n = 16$). (G) Mean number of CaP pMN branches per embryo analysed in the panel-F embryo set. (A, C, F) Arrows and asterisk respectively indicate hyper-branched and truncated ventrally projecting pMN CaP axons. Empty arrowheads indicate normal dorsally projecting pMN MiP axons. Scale bars: 25 μm. (B, D, G) Non-blind quantifications were performed on 24 spinal hemisegments located around the yolk tube per embryo. Analysed larvae were pooled from three independent experiments. Violin Plots; horizontal bars indicate the median ± the 1st and 3rd quartiles. One-way ANOVA test with Bonferroni's post test (D, G) or Kruskal–Wallis ANOVA test with Dunn's post test (B, E). *P* values are displayed on graphs. Source data are available online for this figure.

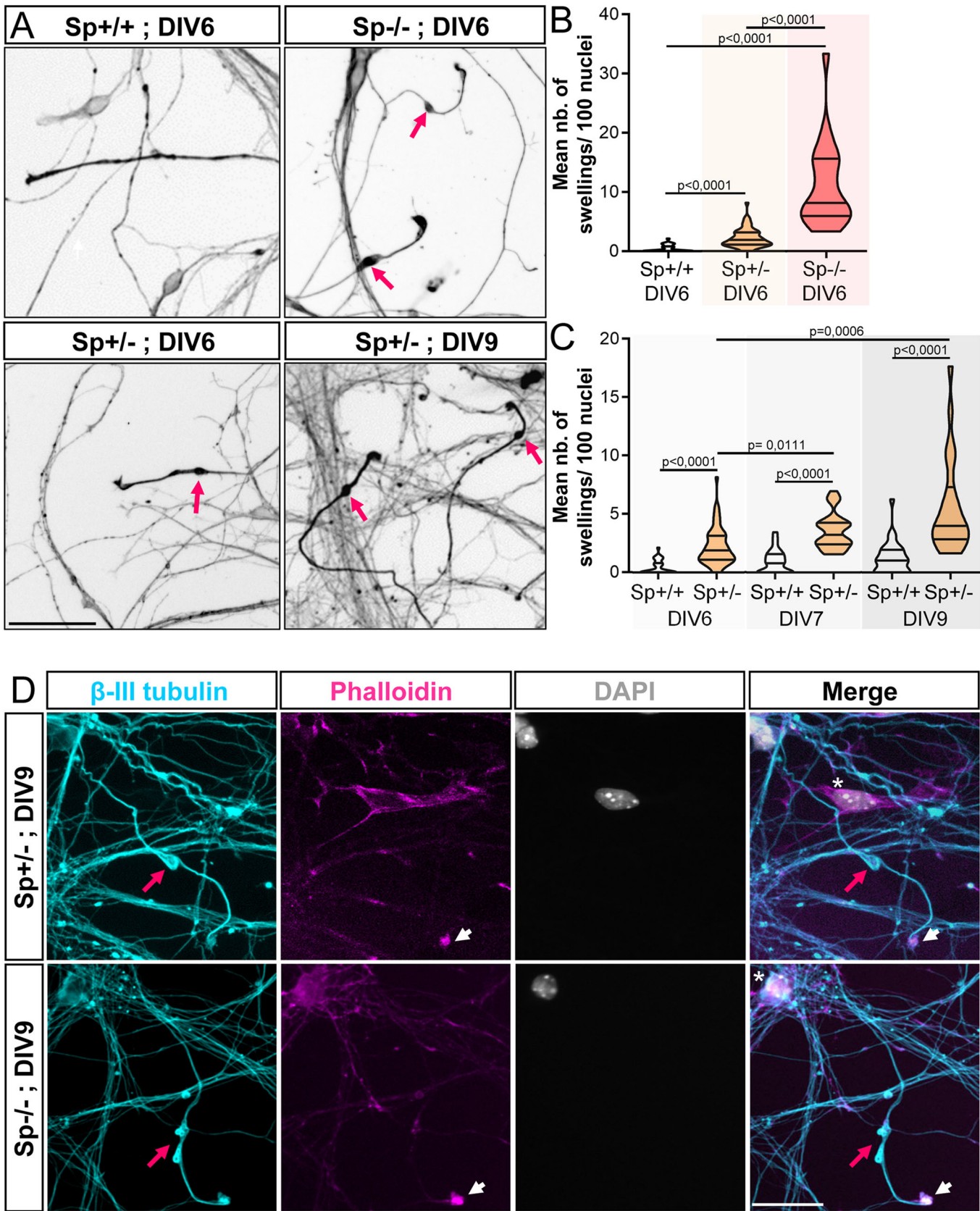

◀ **Figure EV6.   Mouse Sp + /- cortical neurons exhibit a significant number of axonal swellings.**

(A) Mouse Sp + /+, Sp + /- and Sp-/- cultured cortical neurons immunolabelled with a βIII-tubulin antibody at different days in vitro (DIV). Pink arrows point at axonal swellings. Scale bars: 50 μm. (B, C) Mean number of axonal swellings per 100 nuclei. At least 2500 neurons from two independent experiments were analysed in unblind manner per condition. Violin Plots; horizontal bars indicate the median ± the 1st and 3rd quartiles. Kruskal–Wallis ANOVA test with Dunn's post hoc test. $P$ values are displayed on graphs. (D) Primary culture of Sp + /- and Sp-/- cortical neurons immunolabelled at DIV9 with βIII-tubulin antibodies, F-actin probes (Phalloidin, pink) and DAPI (grey). Axonal swellings (pink arrows) of Sp + /- exhibit the same characteristic features as those described in Sp-/- cultures. They are (i) always located close to the growth cone (arrowheads), (ii) their diameter is at least 2 to 3 times larger than the diameter of the axon shaft, (iii) they are always strongly labelled by tubulin antibodies and (iv) are always negative for DAPI staining (asterisk). Scale bars: 25 μm. Source data are available online for this figure.

