## [Peer Review File · The EMBO Journal]

Tubulin glutamylation regulates axon guidance via the selective tuning of microtubule-severing enzymes

Daniel Ten Martin, Nicolas Jardin, Juliette Vouigny, Francois Giudicelli, Laïla Gasmi, Naomi Berbée, Veronique Henriot, Laura Lebrun, Cecile Hautmaitre, Matthias Kneussel, Xavier Nicol, Carsten Janke, Maria Magiera, Jamilé Hazan, and Coralie Fassier

Corresponding author(s): Coralie Fassier (coralie.fassier@inserm.fr) , Maria Magiera (maria.magiera@curie.fr), Jamilé Hazan (jamilé.hazan@college-de-france.fr)

Review Timeline:

Submission Date:	29th Jan 24
Editorial Decision:	22nd Feb 24
Revision Received:	15th Sep 24
Editorial Decision:	15th Oct 24
Revision Received:	20th Oct 24
Accepted:	24th Oct 24

Editor: Ieva Gailite

Transaction Report:

Dear Dr. Fassier,

Thank you for submitting your manuscript to The EMBO Journal. We have now received comments from three reviewers, which are included below for your information.

As you will see from the reports, the reviewers find the topic and the findings interesting. However, reviewers #1 and #2 indicate that further, more direct evidence for changes in microtubule polyglutamylation and microtubule would need to be provided. Furthermore, reviewer #2 requests further insight into the cellular basis of the observed axon pathfinding defects and the possible indirect effects due to potential disruption of primary motor neuron pathfinding.

Based on the interest expressed in particular by reviewers #1 and #3, I would like to invite you to address the comments of the reviewers in a revised version of the manuscript, in particular focussing on the points outlined above. Please note that it is The EMBO Journal policy to allow only a single major round of revision and that it is therefore important to resolve the main concerns at this stage. I think that it would be useful to discuss the revision in more detail via email or phone/videoconferencing - please let me know which option you prefer.

We generally allow three months as standard revision time, which can be extended to six months in the case of major revisions. As a matter of policy, competing manuscripts published during this period will not negatively impact on our assessment of the conceptual advance presented by your study. However, please contact me as soon as possible upon publication of any related work to discuss the appropriate course of action. Should you foresee a problem in meeting this deadline, please let us know in advance to discuss an extension.

When preparing your letter of response to the referees' comments, please bear in mind that this will form part of the Review Process File and will therefore be available online to the community. For more details on our Transparent Editorial Process, please visit our website: <https://www.embopress.org/page/journal/14602075/authorguide#transparentprocess>. Please also see the attached instructions for further guidelines on preparation of the revised manuscript.

Please feel free to contact me if have any further questions regarding the revision. Thank you for the opportunity to consider your work for publication, and I look forward to discussing your revision with you.

Yours sincerely,

Ieva Gailite

- a point-by-point response to the referees' comments, with a detailed description of the changes made (as a word file).

- a word file of the manuscript text.
- individual production quality figure files (one file per figure)
- a complete author checklist, which you can download from our author guidelines (<https://www.embopress.org/page/journal/14602075/authorguide>).
- Expanded View files (replacing Supplementary Information)
Please see out instructions to authors
<https://www.embopress.org/page/journal/14602075/authorguide#expandedview>

We realize that it is difficult to revise to a specific deadline. In the interest of protecting the conceptual advance provided by the work, we recommend a revision within 3 months (22nd May 2024). Please discuss the revision progress ahead of this time with the editor if you require more time to complete the revisions.

Referee #1:

Using a careful and exhaustive combination of loss of function and rescue experiments in zebrafish embryos Ten Martin and colleagues convincingly propose a novel mechanism for axon guidance in motor neuron during early development. They show that it involves the microtubule polyglutamylases TTLL6 and TTLL11. Moreover, these two enzymes have non redundant functions and selectively promote the activity of two microtubule severing enzymes, katanin and spastin respectively. Finally, the authors provide a proof of concept supporting the role of TTLL11 in promoting spastin activity using a mouse model for the spastic paraplegia.

Overall, this is a really interesting work that provides a novel understanding on how microtubules acquire relevant functionalities through very specific post-translational modifications. the manuscript is well written and the experiments are solid and support the conclusions the authors make. There are however a few issues that need to be addressed.

Specific comments:

1. The authors used morpholino based silencing and rescue experiments to test the role of TTLL6 and TTLL11 in axon guidance in motor neurons during zebrafish early development. In a previous work the same experimental approach was used to address the role of TTLL11. The authors described there a role for TTLL11 in chromosome segregation fidelity in the early phases of zebrafish embryonic development, also supported by additional data (Zadra et al, 2022). It is surprising that the authors did not comment on these data. They should be put in the context of their results by discussing whether the defects in cell division during the early embryonic cell cycles in MO TTLL11 (and knockouts) may have (or not) an impact on the phenotypes they describe.
2. The only evidences presented in the manuscript for a role of microtubule polyglutamylation in axon guidance are indirect. They are based on the reported activities of the enzymes in other systems and the IF images shown in Fig.4A in which it is very difficult to identify the specific axons described throughout this work. The authors could perform IFs with the GT- and/or the PolyE antibodies in the same samples as shown in Fig4C that include control and MO-TTLL6 and MO-TTLL11 embryos. Visualizing the (poly)glutamylation signal in motor neurons in these embryos at 72hpf would provide a better connection between this post-translational modification of the microtubules, TTLL6 and TTLL11 activities and the different motor guidance phenotypes.
- 3- The author should provide a better evidence for the role of microtubule polyglutamylation in axon guidance by performing rescue experiments with the catalytic dead enzymes.
4. For the results shown in Fig. 3A and D, the following controls should be included: MOctr+ mRNASpast and MOctr+ mRNAKATNA1 to verify that the overexpression of the each of the two severases has no detrimental impact on motor axon development. These data will reinforce the lack of cross rescue observed in these experiments.
5. In Fig.8C, the percentage of transduced versus non-transduced neurons is not clear and statistics are not provided.

Referee #2:

The manuscript by Martin et al. investigates how a specific MT modification, polyglutamylation, modifies the activity of two MT-severing enzymes, p60-catanin and spastin. The authors study this process in the context of the spinal motor neuron wiring in

the zebrafish model. They also perform a series of experiments in the mammalian neuronal culture. They first show that partial and full loss of function of either p60-katanin or spastin leads to motor neuron phenotypes in the fish. The phenotypes appear to be somewhat specific to various subpopulations of motor neurons innervating distinct aspects of trunk muscle. Then they show that knockdown of tubulin glutamylases TTLL6 and TTLL11 rescue partial (but not complete) loss-of-function of p60-katanin or spastin, respectively. They argue these experiments demonstrate that tubulin glutamylases specifically regulate p60-katanin or spastin activity. Finally, they show that TTLL11 overexpression can rescue an axon swelling phenotype in the mammalian cultured neurons that have only one copy of spastin (similar to a loss of spastin in hereditary spastic paraplegia). Overall, the results are interesting as they connect roles of distinct MT-severing enzymes to specific axonal population. The data also argue that somehow these enzymes can read polyglutamylation patterns generated by two different tubulin glutamylases. While this is interesting, the biggest drawback of this study inference of changes in MT dynamics based on genetic manipulations and then observation of axonal phenotypes. In other words, the study lacks a mechanistic insight into what actually happens to MTs and polyglutamylation when they manipulate levels of p60-catanin and spastin as well as TTLL6 and TTLL11. This reduces the overall impact of this work.

Major comments:

1. Given the fact that the authors have two antibodies recognizing long and short chains of glutamates, they should be able to examine how this particular MT modification changes in TTLL6 and TTLL11 mutants. Can they note any differences in 1) different populations of motor neurons; and 2) various regions of motor neurons (axon shafts versus growth cones)? This may provide a clue as to the specific axonal phenotypes in various mutant.
2. The in situ picture for p60-catanin is not informative. It appears that the gene is not expressed in the posterior spinal cord. Is it expressed later in that region? Can the authors actually show that p60-catanin and spastin are expressed in spinal motor neurons, rather than in a whole spinal cord? Finally, given the abundance of scRNA-seq data sets, the authors can easily ask whether p60-catanin and spastin (as well TTLL6 and TTLL11) are expressed in motor neurons at the time of axon extension.
3. The authors claim that "TTLL6 selectively boosts p60-Katanin activity in motor axons to control their targeting". I just do not see how they can make this conclusion without looking directly at p60-Katanin activity or its downstream effect on microtubules. Same applies to their conclusion that TTLL11 regulating spastin activity.
4. Similar to the point above, the authors concludes "TTLL11 also selectively tunes spastin activity in mammalian cortical neurons". It does seem to rescue axonal swelling phenotypes in this model. But how this is connected to MT dynamics and more specifically to spastin mediated MT dynamics is not clear.
5. Can the author identify cellular bases of the motor axon phenotypes in p60-catanin and spastin knock downs. What is leading to axonal phenotypes? Abnormalities in growth cone dynamics?
6. Primary motor neurons pioneer pathfinding by secondary motor neuron in zebrafish. Is the pathfinding of primary motor neurons normal in p60-catanin and spastin mutants? It appears from movies that CaPs may have an abnormal morphology in these strains. What about RoPs and MiPs? I think this is an important question, because if primary motor neurons are affected then defects in secondary motor neurons are not direct.

Minor comments:

It is hard to understand what is exactly happening in the movies one through four and what the abnormalities are. The authors need to annotate them and point out defects.

I assume that behavioral assays were performed in the morphologically normal looking subset of mutants. Otherwise, given the body curvature, they would not locomote well.

Referee #3:

In this manuscript Martin and colleagues show that katanin plays important roles in zebrafish axon guidance. This was previously also shown for spastin however these seem to function non-redundantly. Furthermore they show an important role for microtubule glutamylation in regulating the activity of both severing enzymes. Importantly they found that the 2 TTLL enzymes responsible for the glutamylation are specific to either of the severing enzymes (although previous work showed similar enzymatic activity for both). This is a surprising and important finding that leaves us wondering how this can be possible. And this suggests that the MT code is even more complex than anticipated. Finally they end by showing that reduced spastin activity in mouse neurons can be rescued by increasing its activity by increasing glutamylation, which may have therapeutic potential. I appreciate the thoroughness of the study; most of the experiments were performed both with mutants as well as with morpholino and the conclusions flow logically from the presented data.

My main point of criticism is the visualization of the data throughout the figures, which makes it harder to read the data. Here are several points the authors could try to change to improve the readability:

- Some data seem the same but is plotted differently (e.g. 1B vs 1G)
- It is much easier to read a bar-plot with the text below the column and not a complex legend (especially not a double legend such as 1G,1I)

- Boxplots are not always easy to interpret (e.g. 2B-D white columns) maybe converting them to dotplots/violinplots would be better

- labeling consistency color use could be improved (e.g. some have titles some have not)

Next to this: for fig 8, it is a bit hard for me to interpret the data. The swelling phenotype in the heterozygous neurons seems to be much more subtle (I would never have picked this up). Is this really the same type of swelling? What this performed blindly? And the use of GFP transduction is not clear also because the swollen axons in the zoom seem to be GFP-. It this is just to visualize a single neuron, maybe just say so move fig 8C to the supplementals.

Other:

- What is Zn-5?

- Fig 1 middle-top image: are both dorsal axons defective? The left one seems ok

- Fig 1F 2nd image has no arrows

- Fig 5 H there is a line below somata

- Fig 8 the top-right zoom in 8A is flipped. And in 8B the "d" is not specified in the legend (although this can be understood)

Response to reviewers (Manuscript #EMBOJ-2024-116734)

We thank the reviewers for their unanimous appreciation of our work and their constructive comments. While the reviewers were all convinced that we have identified a novel key mechanism regulating microtubule (MT) functions in axon guidance, they feel that some of our conclusions regarding the control of MT-severing enzyme activity by TLL glutamylases in navigating axons could be strengthened by additional experiments. These analyses mostly aimed at investigating the influence of TLLs on MT polyglutamylation levels in developing motor axons as well as clarifying the impact of MT-severing enzyme depletion and TLL-mediated rescue on MT dynamics .

Please find below our responses to each of their specific issues.

In the revised version of our manuscript, we now provide compelling evidences supporting **(i)** the contribution of TLL6 and TLL11 to MT polyglutamylation in spinal motor axons and **(ii)** the key role of their catalytic activity (i.e., addition of long glutamate side chains on the C-terminal tail of the tubulin) in the selective regulation of p60-Katanin and Spastin-driven motor axon guidance. Our revised manuscript also includes **(iii)** time-lapse videomicroscopy recordings of MT plus-end dynamics at a single axon scale in live zebrafish larvae revealing that TLL-mediated tubulin polyglutamylation rescues the defects of MT dynamics associated with the partial loss of a MT-severing enzyme (herein, p60-Katanin) as well as **(iv)** additional data regarding the consequences of these different MT-regulatory proteins on primary motor axon development.

The 'Results' and 'Discussion' sections have thus been changed accordingly. We have also modified the visualisation of our data throughout the figures according to reviewer 3's recommendations (colour code, violin plots, legends, etc.).

We are convinced that the reviewing process has considerably strengthened our conclusions and improved the quality of our manuscript.

Referee #1:

Using a careful and exhaustive combination of loss of function and rescue experiments in zebrafish embryos Ten Martin and colleagues convincingly propose a novel mechanism for axon guidance in motor neuron during early development. They show that it involves the microtubule polyglutamylases TLL6 and TLL11. Moreover, these two enzymes have non redundant functions and selectively promote the activity of two microtubule severing enzymes, katanin and spastin respectively. Finally, the authors provide a proof of concept supporting the role of TLL11 in promoting spastin activity using a mouse model for the spastic paraplegia.

Overall, this is a really interesting work that provides a novel understanding on how microtubules acquire relevant functionalities through very specific post-translational modifications. the manuscript is well written and the experiments are solid and support the conclusions the authors make. There are however a few issues that need to be addressed.

Specific comments:

1. The authors used morpholino based silencing and rescue experiments to test the role of TLL6 and TLL11 in axon guidance in motor neurons during zebrafish early development. In a previous work the same experimental approach was used to address the role of TLL11. The authors described there a role for TLL11 in chromosome segregation fidelity in the early phases of zebrafish embryonic development, also supported by additional data (Zadra et al, 2022). It is surprising that the authors did not comment on these data. They should be put in the context of their results by discussing whether the defects in cell division during the early embryonic cell cycles in MO TLL11 (and knockouts) may have (or not) an impact on the phenotypes they describe.

We apologise for this omission and agree with the referee that discussing our results in relation to this study is of interest. In the paper from Zadra et al. (2022), the authors indeed described an early embryonic lethality phenotype associated with defects in chromosome segregation fidelity in both *TLL11* CRISPR/Cas9 KO embryos and **embryos injected at 1-cell stage with 2.4 ng of *TLL11* morpholinos**. The overlapping phenotype between their *TLL11* null mutants and morphants suggests that at this dose of morpholino and with this injection procedure (within the cell of a 1-cell embryo), they may have substantially switched off *TLL11* expression. While we used the same morpholino oligonucleotide (MO), which targets the translation initiation site of *TLL11* mRNA, we preliminarily injected a dilution gradient of this MO **in 2-cell-stage embryos (in the yolk, close to the cells)** as we did for all the morpholinos that we used to knockdown the expression of MT-regulatory proteins (Jardin et al., 2018, Fassier et al., 2018, Atkins et al., 2019). This allowed us to determine the dose that does not affect early embryonic development but leads to moderate morphological (curved tail) and/or locomotor defects at 72hpf. Doing so, we showed that injecting 0.8 pmol (\pm 6.6 ng) of this MO **in the yolk at the 2-cell stage (i)** significantly reduced MT polyglutamylation levels in developing motor axons (see revised Fig 4B-C), **(ii)** induced a curvature of the larval body axis (Fig 4D), which is typical of ciliary mutants, and **(iii)** striking motor axon pathfinding defects (Fig 4D), two phenotypes that were both rescued by the overexpression of mammalian *TLL11* but not *TLL6* (revised Fig 5). Yet, in contrast to the MO-induced knockdown in Zadra et al. (2022), our injection strategy did not lead to any increased lethality in the morphants (i.e., 0-2 % as shown for the controls, versus \pm 45% in Zadra et al., 2022) in more than 10 independent experiments.

Altogether, these results validated the efficiency and specificity of our *TLL11* knockdown strategy. It also showed that this partial depletion of *TLL11* (using a higher dose of MO as that used in Zadra et al., 2020, **but injected in the yolk at a later developmental stage**, hence probably resulting in a lower diffused MO concentration within the cells) **prevented early cell-division defects and lethality**, but reproducibly impaired neuronal circuit wiring.

These technical details, comments and conclusions have been added to the “Results” (page 10, lines 3-6) and “Materials and Methods” (page 23, lines 3-6) sections of the revised manuscript.

2. The only evidences presented in the manuscript for a role of microtubule polyglutamylation in axon guidance are indirect. They are based on the reported activities of the enzymes in other systems and the IF images shown in Fig.4A in which it is very difficult to identify the specific axons described throughout this work.

To facilitate the identification of the different motor axon populations, labelled by GT335 and polyE antibodies, we replaced some images of the revised Fig 4A and added annotations.

The authors could perform IFs with the GT- and/or the PolyE antibodies in the same samples as shown in Fig4C that include control and MO-*TLL6* and MO-*TLL11* embryos. Visualizing the (poly)glutamylation signal in motor neurons in these embryos at 72hpf would provide a better connection between this post-translational modification of the microtubules, *TLL6* and *TLL11* activities and the different motor guidance phenotypes.

We agree with the referee that measuring polyglutamylation signal along sMN axons of 72-hpf control and *TLL6* or *TLL11*-depleted larvae would be important to link MT polyglutamylation levels and axon guidance defects. However, quantifications of PolyE (or GT335) fluorescence intensity in sMN axons could not be carried out *in toto* for the following reasons. First, (i) quantifications cannot be performed at a single axon scale since sMN axons grow in a highly fasciculated tract along the path pioneered by pMN axons. Labelling a single sMN axon within a tract could be achieved by driving the mosaic expression of a fluorescent protein or probe through transient transgenesis – as we showed

for the monitoring of growing MT plus ends (revised Fig 7) –, but this is not applicable here due to the lack of genetically-encoded tools to detect endogenous polyglutamylation levels. (ii) Quantifying and comparing the mean fluorescence intensity of the polyE signal along dorsal or rostral motor nerves of control and TLL-depleted larvae is not an alternative strategy as it cannot be achieved without introducing biases in the polyE values given that the contingent of axons within morphant nerves is affected by axon defasciculation and guidance defects compared to controls.

To bypass these limitations, we opted to evaluate the mean fluorescence intensity of PolyE signal *in vivo* along individual pioneer motor (pMN) axons (here, the ventrally-projecting CaP axons) of 26-hpf control, MOTTLL6- and MOTTLL11-injected embryos, which development is also affected by TLL or MT-severing enzyme deficiency (Fig EV5 and Butler et al., 2010). Using this approach, we showed that the depletion of each glutamylase significantly reduced the level of long glutamate chains in pMN CaP axons (Fig 4B-C), confirming TLL6 and TLL11 knockdown efficiency and the key contribution of these two glutamylases to microtubule polyglutamylation in spinal motor axons.

These results are now included in the ‘Results’ section of the revised manuscript (pages 9-10) and presented in the revised Fig 4, panels B and C.

3- The author should provide a better evidence for the role of microtubule polyglutamylation in axon guidance by performing rescue experiments with the catalytic dead enzymes.

As suggested by the referee, we conducted additional rescue experiments in the zebrafish with the same TLL6 and TLL11 catalytically dead variants (Fig EV4) as these used for rescue experiments in cultured mammalian cortical neurons (revised Fig 9). We here showed that unlike wild-type proteins, TLL6 and TLL11 catalytically dead variants respectively failed to rescue the axon pathfinding defects associated with p60-Katanin and Spastin partial depletion (revised Fig EV4). These results demonstrate that TLL-mediated MT polyglutamylation controls p60-Katanin and Spastin-driven motor axon guidance.

These results are described in page 12 (lines 13-14) for p60-Katanin and page 14 (lines 4-6) for Spastin of the ‘Results’ section of the revised manuscript, and presented in the revised Fig EV4.

4. For the results shown in Fig. 3A and D, the following controls should be included: MOctr+ mRNASpast and MOctr+ mRNAKATNA1 to verify that the overexpression of each of the two severases has no detrimental impact on motor axon development. These data will reinforce the lack of cross rescue observed in these experiments.

To address this point, we conducted gain-of-function experiments and co-injected human *KATNA1* or *SPAST/SPG4* mRNA - at the same dose as that used in corresponding rescue experiments - with control MOs. This analysis revealed that neither p60-katanin or spastin overexpression impacted sMN axon pathfinding under these experimental conditions. These results demonstrate that the lack of cross-rescue effect cannot be attributed to a detrimental effect of spastin or p60-katanin overexpression on sMN axon development.

These results are described in the ‘Results’ section of the revised manuscript (page 8, lines 20-23 & page 9, lines 1-2) and presented in the revised Fig 3.

5. In Fig.8C, the percentage of transduced versus non-transduced neurons is not clear and statistics are not provided.

Since both reviewers 1 & 3 found this panel (Fig 8C) unclear, we removed it from fig 8 (now called Fig 9) and stated in the 'Results' section of the revised manuscript (page 16, lines 4-5) that for each experimental condition, the transduction efficiency was not statistically different and at least equal to 97%. Nevertheless, the raw data and corresponding statistical analysis are provided in the Raw Data Table.

Referee #2:

The manuscript by Martin et al. investigates how a specific MT modification, polyglutamylation, modifies the activity of two MT-severing enzymes, p60-catanin and spastin. The authors study this process in the context of the spinal motor neuron wiring in the zebrafish model. They also perform a series of experiments in the mammalian neuronal culture. They first show that partial and full loss of function of either p60-katanin or spastin leads to motor neuron phenotypes in the fish. The phenotypes appear to be somewhat specific to various subpopulations of motor neurons innervating distinct aspects of trunk muscle. Then they show that knockdown of tubulin glutamylases *TLL6* and *TLL11* rescue partial (but not complete) loss-of-function of p60-katanin or spastin, respectively. They argue these experiments demonstrate that tubulin glutamylases specifically regulate p60-katanin or spastin activity. Finally, they show that *TLL11* overexpression can rescue an axon swelling phenotype in the mammalian cultured neurons that have only one copy of spastin (similar to a loss of spastin in hereditary spastic paraplegia). Overall, the results are interesting as they connect roles of distinct MT-severing enzymes to specific axonal population. The data also argue that somehow these enzymes can read polyglutamylation patterns generated by two different tubulin glutamylases. While this is interesting, the biggest drawback of this study inference of changes in MT dynamics based on genetic manipulations and then observation of axonal phenotypes. In other words, the study lacks a mechanistic insight into what actually happens to MTs and polyglutamylation when they manipulate levels of p60-catanin and spastin as well as *TLL6* and *TLL11*. This reduces the overall impact of this work.

Major comments:

1. Given the fact that the authors have two antibodies recognizing long and short chains of glutamates, they should be able to examine how this particular MT modification changes in *TLL6* and *TLL11* mutants. Can they note any differences in 1) different populations of motor neurons; and 2) various regions of motor neurons (axon shafts versus growth cones)? This may provide a clue as to the specific axonal phenotypes in various mutant.

We fully agree with this comment, which echoed that of reviewer 1. However, as mentioned above, quantifications of polyE signal in control and *TLL6* or *TLL11* morphants could only be carried out in single axons (i.e., the ventrally projecting primary CaP axons) and not in motor nerves (at 72hpf) to avoid any biases linked to the reduced number/contingent of axons within morphant motor nerves (see answer to Reviewer 1, point 2) associated with axon guidance defects. Using this approach, we identified a significant decrease in the mean fluorescence intensity of the polyE signal in ventrally-projecting pMN axons of both *TLL6* and *TLL11* morphant embryos (Fig 4B-C), which happened to be similar in the axon shaft and within the growth cone (see the plot profile in the Extra Figure 1 below). We further established that this reduction in polyE level within pMN axons of *TLL6* or *TLL11* morphants was associated with significant outgrowth and/or branching defects of these CaP axons (revised Fig EV5) providing a correlation between polyE levels and axon pathfinding defects.

Extra figure 1: *TLL6* and *TLL11* knockdown reduces MT polyglutamylation level all along motor axons. Plot profiles showing the distribution of the polyE signal along 6 ventrally-projecting pMN axons of control (MO^{CTL}), *TLL6* (MO^{TLL6}) or *TLL11* (MO^{TLL11}) morphant embryos. Colour-filled areas indicate S.E.M.

This graph could be added to the final version of the manuscript at Reviewer2's discretion.

2. The in situ picture for p60-catanin is not informative. It appears that the gene is not expressed in the posterior spinal cord. Is it expressed later in that region? Can the authors actually show that p60-catanin and spastin are expressed in spinal motor neurons, rather than in a whole spinal cord? Finally, given the abundance of scRNA-seq data sets, the authors can easily ask whether p60-catanin and spastin (as well *TLL6* and *TLL11*) are expressed in motor neurons at the time of axon extension.

To address this point, in collaboration with the transcriptomic facility of the Vision Institute, we first analysed different datasets of zebrafish spinal motor neuron scRNAseq (D'Elia et al., *Cell Rep*, 2023, <https://doi.org/10.1016/j.celrep.2023.113049>; Kelly et al., *eLife* 2023, <https://doi.org/10.7554/eLife.89338.2> and Scott et al., *Dev Biol*, 2021, 10.1016/j.ydbio.2021.07.010). However, in these datasets, the expression of our genes of interest was either low (e.g., spastin) or almost undetectable (e.g., p60-catanin). Nevertheless, it is well-established that scRNA-seq analyses are less sensitive than bulk RNAseq and less exhaustive since they are limited to ± 2000 transcripts, which are often the most abundant ones. To bypass this limitation and the lack of available bulk RNAseq dataset for zebrafish spinal motor neurons, we performed *in situ* Hybridisation Chain Reaction (HCR) experiments in *Tg(Hb9:GFP)* transgenic larvae, expressing the GFP in spinal motor neurons using *spastin* and *p60-katanin* probes. This technique has the advantage of being highly sensitive and allows high-resolution RNA imaging (at a single molecule resolution) in highly autofluorescent samples, like whole-mount zebrafish embryos.

Using this strategy, we showed that *spastin* and *p60-katanin* transcripts are expressed all along the spinal cord of 72-hpf larvae (i.e., the developmental stage at which SMN axon guidance defects were observed), where both transcripts can be detected in spinal motor neurons. Notably, the majority

of spinal motor neurons (GFP+ cells) expressed both *spastin* and *p60-katanin* transcripts (see arrows in our revised Fig EV3B).

These *in situ* data are described page 9 (lines 8-9) of the 'Results' section and presented in the revised Figure EV3B.

3. The authors claim that "TLL6 selectively boosts p60-Katanin activity in motor axons to control their targeting". I just do not see how they can make this conclusion without looking directly at p60-Katanin activity or its downstream effect on microtubules. Same applies to their conclusion that TLL11 regulating spastin activity.

These conclusions have been established on the basis of our *in vivo* axon guidance data and these from past literature ascertaining that TLL-mediated tubulin glutamylation boosts p60-katanin- and spastin-severing activities in *in vitro* reconstituted systems (Lacroix et al., 2010; Vemu et al., 2016, Shin et al., 2019, Genova et al., 2023). However, we acknowledge that due to the lack of direct evidence supporting that TLL6 (or TLL11) rescue the downstream effects of p60-Katanin (or spastin) knockdown on MT dynamics, we should weigh our conclusions. Thus, in this revised manuscript, we now state that "TLL6 selectively influences **p60-katanin-driven motor axon guidance**" to fit with our experimental data (see page 12, line 19) and have modified accordingly our conclusion for TLL11 and spastin (see page 14, line 15).

Furthermore, to address the reviewer concern and strengthen our conclusions, we investigated the cellular mechanisms underlying the rescue effect of these TLLs on the axon guidance defects of MT-severing-depleted neurons. To this end, we monitored MT plus-end dynamics *in vivo* at a single axon scale in live control and *p60-katanin* morphant larvae injected or not with TLL6 mRNA. Because these *in vivo* live imaging experiments are challenging and time consuming, we focused our analysis on the influence of TLL6 on the MT phenotype associated with p60-katanin partial knockdown during the timeframe of the revision process, given that the axon guidance defects caused by *p60-katanin* knockdown is more penetrant than those associated with Spastin depletion. This analysis showed that *p60-katanin* knockdown reduces both MT plus-end density (Fig 7A-C) and growth speed (Fig 7A,D and Movies EV10-12) in these axons. Notably, these defects of MT dynamics, which were consistent with a lack of MT-severing activity as previously described in p60-Katanin and/or Spastin-depleted axons (Butler et al., 2010; Fassier et al., 2013) were rescued by TLL6 overexpression (see revised Fig 7A-D and Movies EV4). Altogether, these results strongly suggest that TLL6 selectively boosts p60-Katanin activity in navigating motor axons to control their targeting.

These results have been included in the revised manuscript (page 13, lines 6-18) and newly added Fig 7 and movies EV4.

4. Similar to the point above, the authors concludes "TLL11 also selectively tunes spastin activity in mammalian cortical neurons". It does seem to rescue axonal swelling phenotypes in this model. But how this is connected to MT dynamics and more specifically to spastin mediated MT dynamics is not clear.

As mentioned in the last paragraph of the 'Results' section, to conduct rescue experiments in mammalian cortical neurons, we focused on the most significant and evolutionarily conserved phenotype arising from defective MT-dynamics in spastin-depleted neurons: the axonal swellings. We have already characterised the cellular events underlying the axonal swellings of spastin KO cortical neurons using different approaches (immunolabelling of tubulin posttranslational modifications,

nocodazole-depolymerisation assays, live imaging of MT plus-end dynamics, electron microscopy, etc.; Tarrade et al., 2006 and Fassier et al., 2013). In these studies, we showed that spastin depletion led to an early (i.e., present before the occurrence of the swelling) and marked impairment of MT dynamics (e.g., reduced number of MT growing plus ends and growth velocity) and structural organisation in cortical neuron axons. Our analysis also revealed that counteracting this defect of MT dynamics using MT-targeting drugs rescues the axonal swelling phenotype (Fassier et al., 2013). Here, we showed that TLL11, shown to boost spastin MT-severing activity in mammalian cells (Lacroix et al., 2010), could also rescue the degenerative hallmark (i.e., swellings) arising from the sustained stabilisation of the MT network caused by spastin haploinsufficiency. Notably, this rescue effect was not observed when we used a catalytically dead variant of TLL11 or when we performed the same experiment in spastin KO neurons. Altogether, these data strongly suggest that TLL11 might selectively boost spastin activity in mammalian cortical neurons. However, for technical limitations summarised below, we could not investigate the rescue effect of TLL11 on MT dynamics in DIV9 Sp^{+/-} neurons. Indeed, the low transfection efficiency required to analyse MT dynamics at a single axon scale in the dense axonal network coupled with the moderate (20%) penetrance of the axonal swelling phenotype in DIV 9 Sp^{+/-} neurons prevented the discrimination between a clear-cut “rescued” axon and a non-swollen axon (for which the defects of MT dynamics may be subtle) and thereby the interpretation of our results.

We have thus toned down our conclusions at the end of the ‘Results’ section by stating that “this beneficial effect of TLL11 on the axonal swelling hallmark of neuronal degeneration was completely lost when the same experiments were conducted in homozygous Sp^{-/-} neurons (Fig 9A-B), suggesting that TLL11 may selectively tune spastin activity in mammalian cortical neurons.” (page 16, lines 7-10).

5. Can the author identify cellular bases of the motor axon phenotypes in p60-Katanin and spastin knock downs. What is leading to axonal phenotypes? Abnormalities in growth cone dynamics?

These relevant points have already been addressed and published by Butler et al. (2010), who used *in vivo* live imaging to show that both *p60-katanin* and *spastin* knockdown **reduced MT dynamics and growth cone motility in 28-hpf zebrafish embryos**. Using similar live imaging approaches we here monitored the outgrowth/navigation of sMN axons, which are the major focus of our studies. It should however be noted that sMN axons are much thinner than pioneer pMN axons and exhibit much smaller growth cones due to their characteristic growth behaviour within a highly fasciculated tract. Based on these features, sMNs thereby offer poor resolution for the imaging of their growth cone structure and dynamic behaviour, even with single axon labelling strategy. Our best high resolution movies of dorsally or rostrally projecting sMN axons (i.e., the two motor neuron populations respectively affected by p60-Katanin and spastin loss of function) are presented in Movies EV1 & EV2 for p60-Katanin and in Jardin et al., 2018 for Spastin. While these videos provided key information about the outgrowth and guidance of these axons, their relevance for clarifying growth cone dynamics remained very limited.

6. Primary motor neurons pioneer pathfinding by secondary motor neuron in zebrafish. Is the pathfinding of primary motor neurons normal in p60-catanin and spastin mutants? It appears from movies that CaPs may have an abnormal morphology in these strains. What about RoPs and MiPs? I think this is an important question, because if primary motor neurons are affected then defects in secondary motor neurons are not direct.

We totally agree with Reviewer 2 about this point since primary CaP motor axons are clearly affected in our videos of *p60-katanin* morphants. Although, pMN data were not included in the submitted version of our manuscript to focus our message on secondary motor axon guidance, we had also analysed the impact of *p60-katanin*, *spastin*, *TLL6* and *11* knockdown on pMN with the same dose of morpholinos as that used to investigate sMN axon guidance defects and conduct rescue experiments. We have shown that ventrally-projecting pMN CaP axons of *p60-katanin* morphants (MO_{Kat}1.3pmol) were abnormally branched (Fig EV5A and Movie EV2) compared to control axons, a phenotype that was rescued by the overexpression of human p60-katanin. In contrast, dorsally projecting pMN MiP axons appeared to develop normally (Fig EV5A,F). These results indicate that the pathfinding defects of dorsally-projecting sMN axons in MO^{Kat/1.3 pmol} morphants (Fig 1A) may not be secondary to developmental defects of the pioneer pMN MiP axons. Consistently, while the partial knockdown of *spastin* (0.2 pmol MO^{Spast}) led to the misrouting of rostrally-projecting sMN axons, it did not impact pMN axons, strengthening the idea that the sMN axon guidance defects we observed are not primarily caused by alterations of the pMN axons. Finally, the hyperbranching phenotype of *p60-katanin* morphant pMN CaP axons was also observed in *TLL6* and *TLL11*-depleted larvae (Fig EV5C-D) but was only rescued by *TLL6* (and not *TLL11*) overexpression (Fig. EV5F-G). These results suggest that *TLL6*-mediated tubulin glutamylation also selectively controls p60-katanin-driven pMN axon development (Fig EV5).

Primary motor neuron data are briefly mentioned in the revised version of the manuscript (page 12, lines 14-18 & page 14, lines 6-9) and are presented in the revised Figure EV5.

Minor comments:

It is hard to understand what is exactly happening in the movies one through four and what the abnormalities are. The authors need to annotate them and point out defects.

We agree with the referee that it is difficult for lay readers to identify the guidance defects on these videos. We have thus changed the layout of the movies, assembling control and morphant videos side by side, and added annotations to indicate axon guidance defects.

I assume that behavioral assays were performed in the morphologically normal looking subset of mutants. Otherwise, given the body curvature, they would not locomote well.

Behavioural assessment of the larvae injected with 1.3 or 3.4 pmol of p60-Katanin MO was performed on the whole slot of injected larvae including both straight and curved larvae. However, it is worth noting that while the penetrance of the curved-tail phenotype is much lower for the population of 1.3-pmol-MO-injected larvae than that of the 3.4-pmol injected larvae (Movies EV3), the severity of the locomotor deficits (i.e., the reduced swimming covered distance and speed), was not significantly different between the two groups of morphants. Notably, the vast majority of MO^{Kat/1.3pmol} or *Katna1*^{-/-}MZ larvae are straight, yet they are almost immotile (Movies EV3) or exhibit a severe locomotor deficit (Fig 2). Overall, these results clearly demonstrate that the locomotor deficit of p60-Katanin-depleted larvae cannot be attributed to their morphological (curve tail) defects but rather arises from their motor circuit wiring defects.

Referee #3:

In this manuscript Martin and colleagues show that katanin plays important roles in zebrafish axon guidance. This was previously also shown for *spastin* however these seem to function non-redundantly. Furthermore they

show an important role for microtubule glutamylation in regulating the activity of both severing enzymes. Importantly they found that the 2 TLL enzymes responsible for the glutamylation are specific to either of the severing enzymes (although previous work showed similar enzymatic activity for both). This is a surprising and important finding that leaves us wondering how this can be possible. And this suggests that the MT code is even more complex than anticipated. Finally they end by showing that reduced spastin activity in mouse neurons can be rescued by increasing its activity by increasing glutamylation, which may have therapeutic potential.

I appreciate the thoroughness of the study; most of the experiments were performed both with mutants as well as with morpholino and the conclusions flow logically from the presented data.

My main point of criticism is the visualization of the data throughout the figures, which makes it harder to read the data. Here are several points the authors could try to change to improve the readability:

- Some data seem the same but is plotted differently (e.g. 1B vs 1G)

The data presented in panels 1B and 1G are not the same and indicate the mean number of dorsal nerve defects observed in *p60-katanin* morphants and mutants (*katna1^{-/-}*), respectively. We apologize if the graph legends were not easily readable and modified them accordingly to avoid confusion. For mutants, we used another type of graph depicting the percentage of fish with dorsal or rostral nerve defects (Fig 1G,I) since as for many KO mutants – versus morphants –, the penetrance of the phenotype is less severe due to genetic compensation effects. We hope that these additional graphs give a better visibility of our data.

- It is much easier to read a bar-plot with the text below the column and not a complex legend (especially not a double legend such as 1G,1I)

- Boxplots are not always easy to interpret (e.g. 2B-D white columns) maybe converting them to dotplots/violinplots would be better

- labelling consistency color use could be improved (e.g. some have titles some have not)

We agree with the referee on these three points and have thus replaced Box and Whisker graphs by Violin plots, moved the figure legend below each violin plot and uniformised the colour code used for the different experimental conditions throughout the figures of the revised manuscript.

Next to this: for fig 8, it is a bit hard for me to interpret the data.

The swelling phenotype in the heterozygous neurons seems to be much more subtle (I would never have picked this up). Is this really the same type of swelling?

Axonal swellings in *Sp^{+/-}* cultures are similar to those observed in *Sp^{-/-}* cultures except that at a similar time point, they are slightly smaller in terms of volume. This is due to the fact that the size of the swellings as well as their frequency increase with time in culture. Nevertheless, it is worth noting that the size of these axonal swellings is not uniform between axons of a same culture either in *Sp^{+/-}* or *Sp^{-/-}* cultures. Thus, since we first described these swellings (Tarrade et al., 2006), we used specific criteria for unambiguously identifying them: (i) they are located in the distal part of the axon at a distance between 5 to 80 μm from the growth cone, which is identified using the phalloidin staining (i.e., labelling of F-actin), (ii) their diameter is at least **two to three times larger** than the diameter of the axon shaft, (iii) they are always strongly labelled by tubulin antibodies and (iv) are always negative for DAPI staining.

To facilitate their identification, we have added a novel panel in the revised figure EV6 (Fig EV6D), showing that the swellings observed in *Sp^{+/-}* cortical neuron cultures meet these different

criteria like Sp^{-/-} cortical neurons. We have also used empty arrowheads to point at the location of the growth cones on the inset panels of the revised Fig 9A.

- What this performed blindly?

Yes, primary cultures of Sp^{+/-} cortical neurons were transduced by Maria Magiera, who attributed a number to each experimental condition. Then, Coralie Fassier, who was the first to describe these swellings in spastin-depleted neurons (Tarrade et al., 2006) and is highly trained to identify them, did all the quantifications blindly. CF was aware of the assignment of each number to an experimental condition by MM only at the time of statistical analyses.

- And the use of GFP transduction is not clear also because the swollen axons in the zoom seem to be GFP-. It this is just to visualize a single neuron, maybe just say so move fig 8C to the supplementals.

Cells were transduced with lentivirus expressing TLL6 and TLL11 cDNAs fused to the GFP. The GFP staining was thus just used to evaluate the transduction efficiency by quantifying the number of GFP-positive somata. This parameter is critical to ensure that the potential lack of rescue effect is not related to a decrease in the transduction efficiency but only to the inability of the overexpressed protein to prevent/delay the formation of axonal swellings for each experimental condition. We here showed that the transduction efficiencies were not statistically different between each experimental condition (see raw data table of Fig 9) and were at least equal to 97%. For clarity, we removed the panel C of the revised Fig 9 (ex-Figure 8) and mentioned the transduction efficiency in the text (page 16, lines 4-5). Furthermore, the data and their corresponding statistical analysis are provided in the Raw Data Table.

The GFP was detected in almost all axons (including swollen axons) but was not always visible on the images (Fig 9). This is due to the fact that the exposure time used for image acquisition was voluntarily reduced to avoid the overexposure of the neuronal somata.

Other:

- What is Zn-5?

Zn-5 is an antibody directed against DM-GRASP/Neuroilin. DM-GRASP is a transmembrane protein with five extracellular immunoglobulin domains, which is expressed by secondary (sMN) but not primary (pMN) motor neurons during zebrafish development. The Zn-5 antibody is thus commonly used to specifically labelled sMN axons in zebrafish larvae. This missing information has been added to the 'Materials and Methods' section of the revised manuscript (page 25, lines 6-10)

- Fig 1 middle-top image: are both dorsal axons defective? The left one seems ok

On this middle-top image, the contingent of axons forming the dorsal nerve is abnormally split into two distinct fascicles: the one on the left seems to take the right path while the one on the right is misrouted and abnormally crosses the lateral myoseptum (i.e., frontier between two somites; dotted line) to reach the neighbouring somite, which is never observed in control larvae. However, it is not uncommon to observe hemisegments where all the dorsally projecting axons are mistargeted like in the upper images of the revised Fig 6A (middle-top image).

- Fig 1F 2nd image has no arrows

Arrows have been added to all images of the revised Fig 1F.

- Fig 5 H there is a line below somata

This line has been removed from the revised Fig 5H.

- Fig 8 the top-right zoom in 8A is flipped. And in 8B the "d" is not specified in the legend (although this can be understood)

The top-right zoom has been flipped back on the revised Fig 9 (ex-Fig 8) and the "d" is now specified in the Fig 9 legend of the revised manuscript (page 53, lines 14-15).

Dear Coralie,

Thank you for submitting a revised version of your manuscript. I sincerely apologise for the protracted assessment process due to delays in referee report submission and the high number of submissions that we receive at the moment. We have now received input from two of the original reviewers, who find that their previous concerns have been addressed satisfactorily.

Therefore, there now remain only a few editorial points that need addressing before I can extend official acceptance of the manuscript:

1. Please submit up to five keywords.
2. Please merge the "Funding" section with "Acknowledgments" and check that the funding information is correct and identical both in the manuscript and our online system. Currently, Ministère de l'Education Nationale, de la Recherche et de la Technologie and the European MARIE SKŁODOWSKA-CURIE Doctoral Network EGRET-AAA are missing in the online system.
3. CRedit has replaced the traditional author contributions section because it offers a systematic, machine-readable author contributions format that allows for more effective research assessment. Please remove the Authors Contributions from the manuscript and use the free text boxes beneath each contributing author's name in our online submission system to add specific details on the author's contribution. More information is available in our guide to authors.
4. Please rename "Competing interests" section into "Disclosure and competing interests statement" (further info: <https://www.embopress.org/page/journal/14602075/authorguide#conflictofinterest>).
5. We require a Data Availability Section at the end of Materials and Methods. As far as I can see, no data deposition in external databases is needed for this paper. If I am correct, then please state in this section: This study includes no data deposited in external repositories. Further information can be found at <https://www.embopress.org/page/journal/14602075/authorguide#dataavailability>
6. Please update movie nomenclature to that of Movie EV1-EV4 and remove movie legends from the manuscript text file.
7. In the reference section, please update the reference to Depienne et al. according to The EMBO Journal format (10 author names followed by et al.).
8. Figure panels for figures 10A-B and EV1A-C are not mentioned in the manuscript text, please add the corresponding callouts.
9. In our standard image integrity check, we noted a reuse of the image panels between figures 4D and 5B (MOCTI). If this is intentional, please indicate in the figure legends that the images are derived from the same experiment/sample.
10. In our standard source data check, we have noted unexplained numerical duplications in the source data for Figure 7C. I have attached the corresponding files with the detected duplications labelled in colour. Please take a look and correct as needed. A brief explanation would be very helpful.
11. Our data editors have flagged the following issues in figure legends that need correcting:
 - Please provide the exact p values in the legends of figures 1b-e, g-i; 2b-c, e-f; 3b-c, e-f; 4c, e-j; 5c-h; 6b-e; 7c-d; 8b-d, f-g; 9b; EV 1b-c; EV 2f-g; EV 4b, d; EV 5b, d-e, g; EV 6b-c.
 - Please note that in figures 2e; 5g; EV 6b-c; there is a mismatch between the annotated p values in the figure legend and in the figure file that should be corrected.
 - Please provide information on the number and nature of replicates in the legends of figures 1b-d, h; 3e-f; 4e-g, i; 6h.
 - Please define the error bar in the legend of figure EV 2g.
12. Papers published in The EMBO Journal are accompanied online by a 'Synopsis' to enhance discoverability of the manuscript. It consists of A) a short (1-2 sentences) summary of the findings and their significance, B) 3-4 bullet points highlighting key results and C) a synopsis image that is 550x300-600 pixels large (width x height, jpeg or png format). You can either show a model or key data in the synopsis image. Please note that the image size is rather small and that text needs to be readable at the final size. Please send us this information together with the revised manuscript.

With best wishes,

leva

leva Gailite, PhD
Senior Scientific Editor
The EMBO Journal
Meyerohofstrasse 1
D-69117 Heidelberg

Tel: +4962218891309
i.gailite@embojournal.org

We realize that it is difficult to revise to a specific deadline. In the interest of protecting the conceptual advance provided by the work, we recommend a revision within 3 months (13th Jan 2025). Please discuss the revision progress ahead of this time with the editor if you require more time to complete the revisions.

Referee #1:

The authors have addressed carefully all the concerns that were raised in the first revision of the manuscript. They provide new data, a revised version of the text and figures as well as detailed and reasoned answers to all the points from the different reviewers. The quality of the work and the conclusion has greatly improved and I am happy to recommend publication.

Referee #2:

This is a revised version of the manuscript by Martin et al. In the original version, my main concern was that the connection between the microtubule (MT) modifying enzymes and the observed axonal phenotypes was not direct. In the revised version, the authors have done a better job of demonstrating that the induced phenotypes are linked to changes in MT modifications and dynamics, such as through the inclusion of EB3 analysis. They also investigated defects in primary versus secondary motor axons. Overall, the revised manuscript is much improved and addresses most of my concerns.

I do have an additional comment: Given the relatively ubiquitous expression of spastin and katanin, why do the authors believe that some spinal motor neurons are affected while others are not? This would be a valuable point to discuss.

10. In our standard source data check, we have noted unexplained numerical duplications in the source data for Figure 7C. I have attached the corresponding files with the detected duplications labelled in colour. Please take a look and correct as needed. A brief explanation would be very helpful.

Many thanks again for having pointed out these errors in the raw data file. We identified the source of the problem (copy/paste problem or forgetting to average two values; see the table below for precise explanations) and corrected the false values. We redid the statistical analysis and updated both the graph (figure 7C) and the Fig 7C sheet in the raw data table.

Exp.1	38,2396		Exp.1	18,17237793		Exp.1	55,7621
Exp.1	37,8121		Exp.1	5,574136009		Exp.1	32,4074
Exp.1	42,4363		Exp.1	10,48883	This value is the good one.	Exp.1	24,1546
Exp.1	341,807		Exp.1	12,51251251		Exp.1	45,4506
Exp.1	43,0098		Exp.1	10,89324619		Exp.1	73,9723
Exp.1	58,3226		Exp.1	63,16555109		Exp.1	64,7523
Exp.1	43,1136		Exp.1	15,9590368		Exp.1	7,89889
Exp.1	29,321		Exp.1	52,68199234		Exp.1	21,645
Exp.1	54,8267		Exp.1	66,3674186		Exp.1	73,6278
Exp.1	45,4358		Exp.1	17,7523694		Exp.1	28,0753
Exp.1	64,3046		Exp.1	9,661835749		Exp.1	32,1502
Exp.1	52,1395		Exp.1	9,29800093		Exp.1	38,9482
Exp.1	70,3849		Exp.1	0		Exp.1	31,3283
Exp.1	51,4728		Exp.1	70,13474456		Exp.1	48,5133
Exp.1	65,7139		Exp.1	21,28565347		Exp.1	23,3645
Exp.1	64,758		Exp.1	7,358351729		Exp.2	31,2989
Exp.1	13,8028		Exp.1	5,201831045		Exp.2	47,1772
Exp.1	15,8278		Exp.1	0		Exp.2	106,061
Exp.1	67,2646		Exp.1	47,13804774		Exp.2	86,4937
Exp.1	117,96		Exp.1	33,51650395		Exp.2	41,0182
Exp.1	126,451		Exp.1	11,0473798		Exp.2	25,9403
Exp.1	17,9937		Exp.1	24,89884843		Exp.2	32,0104
Exp.1	29,1219		Exp.1	12,48439451		Exp.2	49,0196
Exp.1	25,1383		Exp.2	17,15482244		Exp.2	38,4347
Exp.2	153,085		Exp.2	13,36387934		Exp.2	23,5522
Exp.2	83,1146	The mean of these two values should has been done; good value: 77.5770099563718	Exp.2	8,481764207		Exp.2	38,8894 This is the good value
Exp.2	72,0394		Exp.2	22,0459539		Exp.2	32,2073
Exp.2	40,2394	Error copy/paste formula vs value; goodvalue: 27,58573	Exp.2	19,64887496		Exp.4	77,4888
Exp.2	26,5307		Exp.2	35,71428571		Exp.4	63,0964
Exp.2	31,362		Exp.2	6,41025641		Exp.4	31,5657
Exp.2	37,4532		Exp.2	32,21611916		Exp.4	87,4175
Exp.2	48,4713		Exp.2	34,43481774		Exp.4	59,5462
Exp.2	50,7055		Exp.2	25,25252525		Exp.4	40,2576
Exp.2	83,3545		Exp.2	24,07901736		Exp.4	71,1111
Exp.3	138,527		Exp.2	39,15810063		Exp.4	61,3695
Exp.3	64,4841		Exp.2	20,76843198		Exp.4	63,6574
Exp.3	21,1096		Exp.2	42,31845434		Exp.4	81,1209
Exp.3	41,179		Exp.2	26,45502646	The mean of these two green values should have been done; the exsacte value is 40,95240		
Exp.3	29,6296		Exp.2	28,45502646			
Exp.3	35,2183		Exp.2	43,94224733			
Exp.3	53,1046		Exp.2	15,64945227			
Exp.3	57,6495		Exp.3	3,946720214			
Exp.3	46,2363		Exp.3	14,36781609			
Exp.3	61,7284		Exp.3	2,718653083			
Exp.3	72,0185		Exp.3	53,76344086			
Exp.3	95,631		Exp.3	16,33986328			
Exp.3	80,4496	This value is the good one	Exp.3	76,69285447			
Exp.3	54,0424		Exp.3	36,32003432			
Exp.3	37,6788		Exp.3	32,27239684			
Exp.3	35,0683		Exp.3	40,51972406			
Exp.3	29,3586		Exp.3	21,25075896			
Exp.3	47,1698		Exp.3	88,6802953			
Exp.3	84,5146		Exp.3	24,91380263			
Exp.3	56,8182		Exp.3	91,53682558			
Exp.3	80,4496	Error copy/paste; correct value: 100,472813238771	Exp.3	97,24394786			
Exp.3	75,3012		Exp.3	28,9017341			

Dear Coralie,

Thank you for addressing the final editorial points. I am now pleased to inform you that your manuscript has been accepted for publication.

Before we forward your manuscript to the publishers, I would like to suggest minor edits in the manuscript title, abstract and synopsis. I have also written a blurb that will accompany the title of your manuscript on our website. Please take a look at the text below and in the attached manuscript text file and let me know if any further edits are needed.

Title:

Tubulin glutamylation regulates axon guidance via the selective tuning of microtubule-severing enzymes

Blurb:

Specific tubulin glutamylases regulate the activity of p60-katanin and spastin during axon pathfinding in zebrafish motor neurons

Synopsis:

Tubulin post-translational modifications modulate recruitment and activity of microtubule severing proteins. This study shows that specific tubulin glutamylases (TTLLs) control neuronal circuit wiring and homeostasis in zebrafish and mouse by differentially affecting the activity of two microtubule-severing enzymes: p60-Katanin and spastin.

- Two long-chain tubulin glutamylases with similar biochemical activity in vitro, TTLL6 and TTLL11, have distinct physiological roles in neuronal circuit wiring.
- TTLL11 specifically promotes spastin-dependent microtubule severing and rostral motor nerve pathfinding in zebrafish larvae.
- TTLL6 selectively promotes p60-Katanin activity and dorsal motor nerve pathfinding in zebrafish larvae.
- TTLL11-mediated microtubule polyglutamylation rescues the degenerative hallmarks of hereditary spastic paraplegia associated with spastin haploinsufficiency in mouse cortical neurons.

If you have any questions, please do not hesitate to contact the Editorial Office. Thank you for this contribution to The EMBO Journal and congratulations on a nice study!

Best wishes,

Ieva

>>> Please note that it is The EMBO Journal policy for the transcript of the editorial process (containing referee reports and your response letter) to be published as an online supplement to each paper. If you do NOT want this, you will need to inform the Editorial Office via email immediately. More information is available here: <https://www.embopress.org/transparent->

process#Review_Process